# Analysis of the antimicrobial activity of zinc oxide nanoparticles against drug-resistant bacteria and their applications in the disinfection process

Dalia Alqaffaf[1], Ali M. Atoom[2]*, Rana Abu Huwaij[3], Mai Abdel Haleem A. Abusalah[2], Bayan Tayseer Alzubi[2], Awatef Al-Kaabneh[4], Manal Abdel Haleem A. Abusalah[5], Maher A. Sughayer[1]

1 Department of Pathology and Laboratory Medicine, King Hussein Cancer Center, Amman, Jordan,
2 Department of Medical Laboratory Sciences, Faculty of Allied Medical Sciences, Al-Ahliyya Amman University, Amman, Jordan, 3 A College of Pharmacy, Amman Arab University, Amman, Jordan,
4 Princess Iman Center for Research and Laboratory Sciences, Jordanian Royal Medical Services, Amman, Jordan, 5 Department of Medical Microbiology & Parasitology, School of Medical Sciences, Universiti Sains Malaysia, Kubang Kerian, Kelantan, Malaysia

* a.atoom@ammanu.edu.jo

## Abstract

### Background

The rise in antimicrobial resistance has necessitated the exploration of novel antimicrobial agents. Zinc oxide nanoparticles (ZnO-NPs) have gained prominence due to their biocompatibility, minimal toxicity, and potent antimicrobial properties. This study evaluates the antimicrobial activity of ZnO-NPs synthesized from *Phoenix dactylifera* root extract and their potential applications in disinfection.

### Methodology

ZnO-NPs were synthesized using an eco-friendly method involving *Phoenix dactylifera* root extract and zinc acetate at varied concentrations and ratios, followed by annealing. The nanoparticles were characterized and assessed for antimicrobial activity against a spectrum of bacterial and fungal isolates using microtiter broth dilution, disc diffusion, and pour plate assays. Disinfection efficacy was evaluated on water samples and surfaces. Additionally, the impact of ZnO-NPs on acid-fast bacilli (AFB) isolates was examined using VersaTrek Mycobottles.

### Results

ZnO-NPs exhibited potent antimicrobial activity against Gram-positive and Gram-negative bacteria, with minimum inhibitory concentrations (MICs) ranging from 9.7 to 310 μg/mL. Disc diffusion assays revealed larger inhibition zones in Gram-positive bacteria compared to Gram-negative strains, with MRSA showing the widest zone

**Data availability statement:** All data are in the manuscript.

**Funding:** This work was funded by School of graduate studies, Al-Ahliyya Amman University, Amman, Jordan (Grant number 12/11/ 2020-2021). In addition, microbial isolates were collected from King Hussein Cancer center, Amman, Jordan (IRB approval number 21 KHCC 047). The funder had no role in study design, data collection and analysis, decision to publish, or preparation of the manuscript. Dates removed for consistency.

**Competing interests:** The authors have declared that no competing interests exist.

(24 mm). ZnO-NPs significantly reduced colony-forming units (CFU) on water and surfaces, achieving complete bacterial inhibition on laboratory benches within 20 minutes. The nanoparticles demonstrated bactericidal effects against AFB isolates, highlighting their broad-spectrum efficacy.

## Conclusion

The study underscores the potential of ZnO-NPs as a versatile antimicrobial agent effective against MDR (multidrug-resistant) pathogens and environmental contaminants. Their rapid action and broad-spectrum activity make them suitable for disinfection in clinical and environmental settings. Future research could explore synergistic combinations with traditional antimicrobials to enhance efficacy against infections that are resistant to treatment.

## Introduction

The development and application of nanotechnology have revolutionized various scientific fields, including medicine, agriculture, biology, engineering, pharmacy, the textiles and food industries, and environmental science [1,2]. Nanoparticles have garnered significant attention due to their unique physicochemical properties and potential to address pressing issues such as antimicrobial resistance [2–4]. The rapid rise of antibiotic-resistant pathogens has necessitated the exploration of novel antimicrobial agents [5]. One promising candidate is zinc oxide nanoparticles (ZnO-NPs), which have garnered significant attention due to their biocompatibility, minimal toxicity, excellent chemical stability, wound healing, and antibacterial and antifungal properties [6]. Zinc oxide nanoparticles are renowned for their broad-spectrum antimicrobial activity, which effectively targets a wide range of pathogenic microorganisms, including both Gram-positive and Gram-negative bacteria, fungi, and viruses [4,7,8]. Studies have demonstrated that ZnO-NPs possess significant antibacterial properties against pathogens such as *Candida albicans*, *Aspergillus niger, Streptococcus pyogenes, S. aureus, Acinetobacter baumannii*, methicillin-resistant *Staphylococcus aureus* (MRSA), *Bacillus subtilis, Pseudomonas aeruginosa, Mycobacterium tuberculosis, Bacillus megaterium, Escherichia coli,* and *Klebsiella pneumonia*, especially multi-drug resistant (MDR) pathogens including ESBL-producing strains [9–15]. Their small size and large surface area enhance their interaction with microbial cells, leading to effective antimicrobial action through multiple mechanisms such as the generation of reactive oxygen species (ROS), disruption of cell membranes, and the release of zinc ions, which can penetrate microbial cells, disrupt various enzymatic systems and interfere with metabolic processes critical for cell survival [3,4,16,17]. Several studies have shown that ZnO-NPs can effectively reduce bacterial colony-forming units (CFUs), making them suitable for environmental and clinical disinfection [16,18–20]. For instance, ZnO-NPs have been successfully incorporated into wound dressings, preventing infections and promoting wound healing [21]. The likelihood of infection during the healing process has been demonstrated to be reduced by ZnO

NPs embedded in collagen dressings [22], chitosan hydrogel [23], or cellulose sheets [24]. In water treatment, ZnO-NPs have been employed to disinfect water by targeting and eliminating waterborne pathogens, thus reducing the risk of water-borne diseases [16,18]. The study by Wang, Hu, and Shao (2017) highlighted the antimicrobial potential of nanoparticles. Similarly, ZnO-NPs have shown promise in food preservation by being incorporated into polymeric matrices to enhance packaging material properties and provide antimicrobial activity [25]. Recent studies by Ullah et al. (2024) have shown that ZnO-NPs synthesized using plant extracts, such as wild olive leaf extract, exhibit high catalytic performance and remarkable selectivity (up to 80%) for hydrogen generation from the formic acid/sodium formate system [26].

The study by Ghaffar et al. (2023) demonstrated that ZnO nanoparticles synthesized using olive fruit extract showed excellent photocatalytic and antioxidant activities, highlighting their potential as sustainable and eco-friendly agents for wastewater treatment [27]. Building on this, the present study explores a novel plant source, *Phoenix dactylifera* root extract, for ZnO nanoparticle synthesis, further advancing eco-friendly nanotechnology with practical antimicrobial and disinfection applications. Abbas et al. (2023) demonstrated that ZnO-based nanostructures, particularly surface-functionalized nano-flakes, significantly enhance sensitivity and selectivity in sensing applications due to their large surface area and effective interaction with target molecules [28]. Extending this concept, the present study highlights the antimicrobial versatility of green-synthesized ZnO nanoparticles, emphasizing their effective interaction with multidrug-resistant pathogens and real-world contaminated surfaces. Green-synthesized ZnO nanoparticles using *Equisetum diffusum* D extract have demonstrated multifunctional potential, including effective photocatalytic degradation of various dyes and notable antimicrobial properties.

The study by Assad et al. (2023) highlights the growing relevance of phyto-functionalized ZnO NPs in environmental and biomedical applications [29]. The study by Siddique et al. (2024) underscores the significance of plant-mediated ZnO NPs in environmental remediation and controlling drug-resistant bacteria [30]. Meanwhile, the novelty of the present study lies in the green synthesis of ZnO-NPs using *Phoenix dactylifera* root extract, an underexplored natural resource as a reducing agent. Additionally, it uniquely evaluates the disinfection efficacy of these nanoparticles on real-world environmental surfaces and water samples, bridging gaps left by previous research.

Moreover, the synthesis and optimization of ZnO nanoparticles (NPs) have evolved to include green and eco-friendly methods. Biological substrates, such as fungus, plant extracts, yeasts, phages, algae, and bacteria, are increasingly used to produce ZnO-NPs, offering a sustainable alternative to traditional chemical and physical synthesis methods [4,31,32]. Additionally, this approach also enhances the biocompatibility and safety of the nanoparticles, making them more suitable for biomedical applications [4,32,33]. The studies by Jin and Jin (2021) and Rosli et al. (2021) have emphasized the potential of ZnO-NPs as antimicrobial agents; however, ongoing research into their synthesis, mechanisms, and real-world disinfection applications continues to highlight their versatility and effectiveness [4,33].

The present study addresses this gap by applying green-synthesized ZnO-NPs from *Phoenix dactylifera* roots in actual surface and water disinfection, providing practical antimicrobial insight. The novelty of this study lies in the green synthesis of ZnO-NPs using *Phoenix dactylifera* root extract, an underexplored natural resource, and their comprehensive evaluation against a broad spectrum of clinically relevant and multidrug-resistant bacterial and fungal isolates. Additionally, the study uniquely assesses their disinfection efficacy on actual environmental surfaces and water samples, which is rarely reported in prior research. Therefore, this study aims to synthesize zinc oxide nanoparticles using *Phoenix dactylifera* root extract and comprehensively evaluate their antimicrobial efficacy and potential applications to disinfect water, surfaces, and clinically relevant microbial isolates.

## Methods

### Culture and identification of bacterial and fungal isolates

Bacterial and fungal isolates were retrieved from the King Hussein Cancer Centre (KHCC), with sample collection and research approved by the Ethics Committee of the Institutional Review Board (IRB; approval number 21 KHCC 047).

Samples were collected starting from March 24, 2021. The studied bacterial and fungal isolates included references from the American Type Culture Collection (ATCC) and isolates identified using CAP Proficiency Testing (CAP PT), as detailed in Table 1. Selected bacterial strains are routinely collected in the microbiology laboratory and stored at −70°C for quality control purposes. The current study focuses on a descriptive analysis of these bacterial isolates. As no patient-related data were collected, ethical approval was not required. This study was conducted as basic science laboratory work, and informed consent was not necessary since there was no direct interaction with patients and no collection of personal information. The ATCC strains listed in Table 1, including *E. coli* ATCC 25922, *S. aureus* ATCC 25923, and *P. aeruginosa* ATCC 27853, *Candida albicans* ATCC 90028, *Candida parapsilosis* ATCC 22019, and *Candida glabrata* ATCC 2001, and *Haemophilus influenzae* ATCC 49247 were used as standard quality control strains in the antimicrobial susceptibility assays.

The stock cultures of the microbial isolates (fungi and bacteria) were prepared by mixing 500 µl of a confirmed culture containing either fungi, Gram-negative bacteria, or Gram-positive isolates with 50% glycerol (Sigma-Aldrich, Germany). These were then stored in cryovials (Thermo Fisher Scientific Nalgene Cryovials) at −70°C until further use. Working

**Table 1. list of bacterial and fungal isolates used in this study.**

| Isolate | Source | No. of isolates | Site of sample |
|---|---|---|---|
| **ESBL positive *Enterobacterales*** | Patient isolates | 30 | Blood, urine, wound, PAS |
| | ATCC strains | – | *K. pneumoniae* ATCC 700603 |
| **ESBL negative *Enterobacterales*** | Patient isolates | 30 | Blood, urine, wound |
| | ATCC strains | – | *E. coli* ATCC 25922 |
| ***Pseudomonas aeruginosa* CSPA** | Patient isolates | 20 | Blood, wound, PAS, NS |
| | ATCC strains | – | *P. aeruginosa* ATCC 27853 |
| ***Pseudomonas aeruginosa* CRPA** | Patient isolates | 30 | Wound, PAS, NS, Trap |
| ***Acinetobacter* sp. Multisensitive** | Patient isolates | 40 | Blood, wound, PAS, NS, Trap |
| | ATCC strains | – | ATCC 27244 |
| ***Acinetobacter baumannii* MDR** | Patient isolates | 40 | Blood, wound, PAS, NS, Trap |
| ***Staphylococcus aureus* MSSA** | Patient isolates | 20 | Blood, wound, NS, Trap |
| | ATCC strains | – | ATCC 29213 |
| ***Staphylococcus aureus* MRSA** | Patient isolates | 20 | Blood, NS, Trap |
| | ATCC strains | – | ATCC 25923 |
| **Vancomycin sensitive *Enterococcus faecium*** | Patient isolates | 15 | Urine, PAS |
| **Vancomycin resistant *Enterococcus faecium*** | Patient isolates | 15 | PAS |
| ***Streptococcus pneumoniae*** | ATCC strains | – | ATCC 49619 |
| ***Candida albicans*** | Patient isolates | 20 | TRAP, HVS |
| | ATCC strains | – | ATCC 90028 |
| ***Candida parapsilosis*** | ATCC strains | – | ATCC 22019 |
| ***Candida glabrata*** | ATCC strains | – | ATCC 2001 |
| ***Aspergillus* spp.** | Patient isolates | 10 | Trap, BAL, Sputum |
| **Acid Fast Bacilli (*Mycobacterium tuberculosis*, and *Mycobacterium intracellulare*)** | CAP proficiency testing confirmed isolates. | 2 | E1-03 2021, E1-04 2021 |
| ***Vibrio cholera*** | Patient isolate | 1 | Blood |
| ***Hemophilus influenzae*** | ATCC strains | – | ATCC 49247 |

Footnote: Perianal swab, PAS; Nasal swab, NS; Broncho-alveolar lavage, BAL; American Type Culture Collection, ATCC; College of American Pathologists, CAP; Extended-spectrum beta-lactamases, ESBLs; Multidrug-resistant bacteria, MDR; Carbapenem-resistant *Pseudomonas aeruginosa*, CRPA; Methicillin-resistant *Staphylococcus aureus*, MRSA; Methicillin-susceptible *Staphylococcus aureus*, MSSA.

cultures were established by inoculating a loopful of the stock culture into nutrient broth and incubating it at 37°C for 24 hours.

## Preparation *Phoenix dactylifera* Extract

The roots of *Phoenix dactylifera*, obtained from a juvenile date palm tree at Al-Ahliyya Amman University campus in Amman-Jordan, were used to prepare the plant extract for the green synthesis of ZnO-NPs. The botanical identification of *Phoenix dactylifera* was verified by an expert botanist in the Faculty of Agricultural Technology at the university. The extraction procedure of root fibres was adapted from previously published report by Naser et al., 2021 [34]. Briefly, a knife was used to remove the root hairs lying on the soil's surface. Then the samples were rinsed three times with distilled water and twice with 70% ethanol, dried in an oven (Thermo Fisher, USA) at 40 °C, and finely ground into a fine powder using an 80 mm mesh size grinder. Subsequently, 1 g of the powder was added into a flask with a capacity of 250 mL, containing 100 mL of distilled water, and boiled at 100 °C for 30 minutes. The plant root broth solution was then filtered using filter paper with a pore size of 0.05 mm to eradicate larger particles and stored at 4 °C for further analysis.

## Zinc oxide nanoparticles synthesis

The root extract of *Phoenix dactylifera* was added to three varying concentrations (0.01, 0.1, and 0.6 M) of a dehydrating zinc acetate solution ($Zn(CH_3COO)_2.2H_2O$) in an aqueous bath system. The mixture was continuously stirred at 80 °C for 2 hours. The zinc acetate (Chemsolute,

Germany) and root hair extract mixture were prepared using two distinct volume ratios, which are 1:2 and 1:3. The resultant mixtures were stirred for 2 hours at 80 °C on a hot plate (Thermo Fisher, USA). Followed by filtration using a 0.05 mm filter paper and then cooled down for one hour at room temperature. Subsequently, the white powder was dried at 100°C before its annealing at 450°C for 3 hours. Finally, the resulting powder was measured and stored in closed containers for further characterization and applications. The percentage yield was determined using the following equation [35].

$$= Yield\ \% = \left( \frac{Weight\ of\ obtained\ ZnO - NPs\ after\ drying}{Weight\ of\ dehydrating\ Zinc\ acetat} \times 100\% \right)$$

Where ZnO-NPs concentration used was 0.01g/mL, and the ratio was 1:2.

The ZnO-NPs utilized in this study were synthesized and thoroughly characterized in previous work by our group, Naser et al. (2021), Abu-Huwaij et al. (2022), and Obaid et al. (2023), including detailed analyses such as UV–Vis spectroscopy (UV/VIS-002, Shimadzu, Japan), Fourier-transform infrared spectroscopy (FTIR) (IR Prestige 21, Shimadzu, Japan), X-ray diffraction (XRD) (Riguka, Japan), and scanning electron microscopy (SEM) (Inspect F50-FEI, USA) [34,36,37]. This study is part of an ongoing research series of Prof. Rana Abu-Huwaij, with this being the sixth publication on zinc oxide nanoparticles. It builds on five previously published studies by our research group focused on zinc oxide nanoparticles, further advancing the characterization and applications of these materials [34,36–39]. This characterization included UV–Visible spectroscopy to confirm nanoparticle formation, FTIR to identify functional groups responsible for reduction and stabilization, XRD to determine crystalline structure, SEM for surface morphology, and dynamic light scattering (DLS) for size distribution and polydispersity index (PDI) assessment. Since this study employed the same batch of ZnO-NPs, repeating these analyses would be redundant. However, all experimental protocols and data are available in the referenced publications. The optimal synthesis conditions for ZnO-NPs were achieved using a 1:1 volume ratio of *Phoenix dactylifera* root extract to zinc acetate solution. Under these conditions, the nanoparticles exhibited an average hydrodynamic diameter of 22.57 ± 4.79 nm with a PDI of 0.53 ± 0.02, indicating a moderately uniform particle size distribution. The zeta potential was measured at −19.23 ± 1.40 mV, reflecting a moderately high negative surface charge. This

value suggests good colloidal stability of the ZnO-NPs in suspension, as the electrostatic repulsion between particles helps prevent aggregation and supports their suitability for biological applications.

## The antimicrobial activity of nanoparticles (MIC)

The antifungal activity of synthesized ZnO-NPs was assessed against fungi, Gram-negative bacteria, and Gram-positive bacteria, as shown in the previous "Table 1". The present study employed various techniques to evaluate the antibacterial properties of the synthesized ZnO-NPs. In addition to MIC assays, the disc diffusion method was used to visually validate the antimicrobial potential of ZnO-NPs and facilitate comparison with standard antibiotic susceptibility patterns, thereby providing complementary data on inhibition zone diameters.

## MIC determination by the microtiter broth dilution method

To determine the MICs, the broth microdilution method, a type of liquid bacterial culture assay, was conducted according to the Clinical and Laboratory Standards Institute (CLSI)/NCCLS M07 and M27 [40,41]. This liquid-based approach is widely considered a reliable and quantitative method for evaluating antibacterial activity under controlled conditions, offering greater precision than solid media-based methods like disc diffusion. Before antimicrobial evaluation, microbial isolates were subcultured on fresh Muller Hinton broth (MHB) (Oxoid, UK) and incubated at 37 °C for 24–48 hours. The antimicrobial efficacy of the synthesized ZnO-NPs was evaluated in the presence of approximately $1.5 \times 10^8$ CFU/mL as a final microorganism concentration. This test was conducted using disposable microtitration 96-well plates (Globe Scientific, USA). Serial dilutions of ZnO-NPs determined the MICs. The dilutions were prepared using standard concentrations, with an initial concentration of 0.01 g/mL and a final concentration ranging from $5 \times 10^3$ to 2.4 µg/mL, selected based on prior literature and preliminary inhibitory screening. The wells were inoculated with 50 µL of the ZnO-NPs suspension and 50 µL of overnight-grown test cultures. The negative control (sterile, uninoculated media) and the positive control (50 µL of microbial inoculum only) were also included in separate wells. The plates were incubated at 37 °C overnight under aerobic conditions. The wells were then examined for any colour change, as shown in "Fig 1".

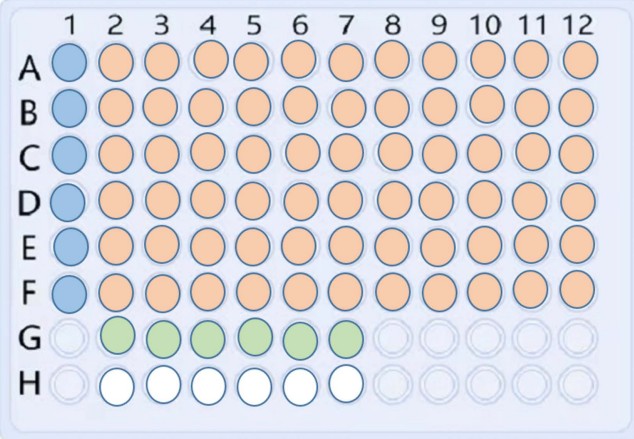

**Fig 1. Microtiter Broth Dilution Method Plate Setup: The figure shows a 96-microtiter plate that represents the experiment setup to evaluate the antimicrobial activity of ZnO-NPs.** Column 1 (A1 to F1 highlighted in blue) was loaded with ZnO-NPs suspension to observe the original solution turbidity. Wells highlighted in orange represent serial double fold dilution of ZnO-NPs in each row, followed by loading each well with standard bacterial suspension to evaluate the effect of ZnO-NPs on different microbial growth through measuring the MIC values. Wells in green at row G, and wells in white at row H were loaded with microbial suspension or media alone to serve as positive and negative controls, respectively.

## Disc diffusion assay

The disc diffusion test was conducted to evaluate the antimicrobial activity of synthesized ZnO-NPs. A concentration of 500 μg/disc of ZnO-NPs-impregnated filter paper discs (Oxoid, UK) was tested against microbial isolate strains, including fungi and bacteria. To prepare the inocula, the bacterial and fungal isolates were cultured on 5 ml of MHB (Oxoid, UK) and incubated overnight. To achieve a turbidity of 0.5 McFarland standards, the microbial concentration was adjusted to approximately $1.5 \times 10^8$ CFU/mL in the microbial suspension. The bacterial and fungal inoculum were spread over the surface of Muller-Hinton agar (MHA) and Sabouraud dextrose agar (SDA) (Oxoid, UK) media, respectively. Subsequently, the pre-prepared discs containing ZnO nanoparticles were used in conjunction with commercially available antibiotic discs for comparison. Discs containing only Zinc acetate were used as a diluent control. The control was prepared by adding μL50 μL of a 0.1317 g/mL (0.6 M) zinc acetate solution with a volume ratio of 1:2. The plates were subsequently incubated at 37°C for 24–48 hours. The sensitivity was assessed by measuring the growth inhibition zone (in millimetres) surrounding the discs. The CLSI was employed to interpret the results [42–45]. While the disc diffusion assay provides initial insight into antibacterial activity, it may not fully reflect real-world conditions. Therefore, to validate and complement the findings, additional experimental approaches, including MIC determination, pour plate assays, and VersaTrek Mycobottle (Thermo Scientific/ Fisher Scientific, USA) evaluations, were also conducted in this study.

## Pour plate assay

The pour plate assay was conducted following the procedure outlined by Cappuccino and Welsh [46]. This study employed the pour plate method to evaluate the effectiveness of ZnO-NPs as a disinfectant for various surfaces (glass, stainless steel, and bench) and water. The principle of this method is to quantify the reduction in the number of CFU of bacterial inoculum in a liquid specimen after a given period of exposure to an antimicrobial agent compared to an untreated specimen.

Glass slides (Globe Scientific, USA), stainless-steel surgical blades (Globe Scientific, USA), and water samples were contaminated with bacterial suspension containing a mixture of (*E. coli, K. pneumoniae, S. aureus, E. faecium*) with and without ZnO-NPs for a duration that gradually increased (5, 10, 15, 20, 25, 30, 60 minutes, and overnight). Swab samples at different time intervals from the glass slides and blades were collected and resuspended in 0.1 mL of sterile normal saline. Additionally, 0.1 mL samples of the contaminated water at different time intervals were used for the pour-plate method. Each 0.1 ml sample was dispensed into the centre of a sterile, empty petri dish (Corning, USA). After that, a tube containing 20 mL of melted agar at 55°C was poured into the petri plate containing the inoculum and thoroughly mixed. After the agar solidified, the plate was incubated at 37°C in ambient air for 24 hours.

This method has been applied to conduct viable plate counts, which involve enumerating the total number of CFU on the surface of the agar and within the agar for a single plate. The inhibition percentage is determined by comparing the count of CFU/mL in the treated sample to that of the untreated sample. A plate appropriate for counting must contain a minimum of 30 colonies and a maximum of 300 colonies. The CFU/mL was calculated using the following formula [47]:

$$CFU/ml = CFU \times dilution\ factor \times 1/aliquot (aliquot\ is\ 0.1 ml).$$

Consequently, the CFU/mL resulting from testing each surface type and water was determined before and after exposure to ZnO-NPs treatments at various time intervals. To quantitatively determine the observed effect, the percentages of decreases in CFU/mL between samples treated with ZnO-NPs and those untreated were calculated.

## Evaluating the disinfectant properties of ZnO-NPs using VersaTrek Mycobottles

VersaTrek Mycobottles (Thermo Scientific/ Fisher Scientific, USA) were employed to determine the impact of ZnO-NPs on acid-fast bacillus bacteria (AFB), specifically whether they have a bactericidal or bacteriostatic effect. Sputum specimens

were obtained from CAP PT, specifically from CAP-PT E1-03 and E1-04 2021.VersaTREK Myco bottles contain growth supplements (GS) to promote the growth of FAB and PVNA (polymyxin B, vancomycin, nalidixic acid, amphotericin B) antibiotic mixtures, as well as other antibiotic supplements (AS) to inhibit any other normal flora in respiratory secretion. Furthermore, each bottle was supplemented with an identical volume of ZnO-NPs (1 mL at 0.01 g/mL). The diluent control bottle was also supplemented with 1 mL of 0.1317 g/mL zinc acetate to determine whether the observed antimicrobial effect was specifically due to ZnO-NPs or the precursor compound (zinc acetate) alone. This control ensures that any observed inhibitory activity is attributed solely to the synthesized nanoparticles (Fig 2). The sputum specimens were treated with equal volumes of NaOH and then mixed directly. The NaOH exposure continued for 15–20 minutes at room temperature, inhibiting all microorganisms present in the specimens without causing any damage to AFB. The solution was centrifuged at 4000 rpm for 20 minutes following phosphate-buffered saline (PBS) buffer to re-regulate the pH. The supernatant was discarded, and the sediment was kept for further analysis. This method was conducted according to the CLSI [48].

## Results

### MIC determination by microtiter broth dilution method

The MIC results indicated that a ZnO-NPs suspension with a concentration of 9.7 µg/mL had a notable inhibitory effect on the vancomycin-sensitive phenotype of *E. faecium* (*E. faecium* VS), *S. pneumoniae*, and *Brucella* spp. On the other hand, the maximum concentration of 310 µg/mL exhibited inhibitory effects on the *P. aeruginosa* CRPA phenotype, as shown in "Fig 3". The MIC values for the remaining microorganisms that were tested are shown in "Tables 2 and S1".

### Antimicrobial activity of ZnO-NPs against different Gram-positive and Gram-negative bacterial isolates

The disc diffusion method was performed to evaluate ZnO-NPs antimicrobial activity against different Gram – positive and Gram – negative bacterial isolates. The findings of this study revealed that **Gram-negative** bacteria showed the widest inhibition zone of 19.6 mm against *E. coli* and *K. pneumoniae* ESBL negative phenotype isolates. On the other hand, the narrowest zone of inhibition measured was 13.0 mm against *P. aeruginosa* CSPA phenotype isolate. Furthermore, the

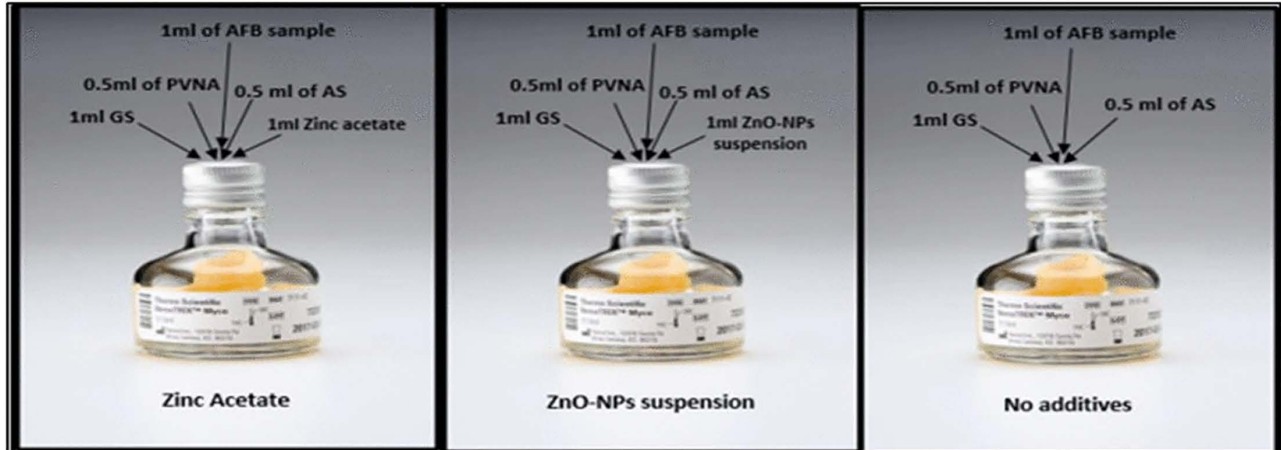

**Fig 2. Schematic representation of VersaTrek Myco bottles used to evaluate the disinfectant properties of ZnO-NPs against acid-fast bacilli (AFB).** Two different CAP proficiency isolates of Mycobacterium tuberculosis were separately inoculated into Myco bottles. GS refers to growth supplements that support AFB growth. A mixture of antibiotics—polymyxin B, vancomycin, nalidixic acid, and amphotericin B (PVNA)—along with an additional antibiotic supplement (AS), was used to inhibit the growth of bacterial contaminants.

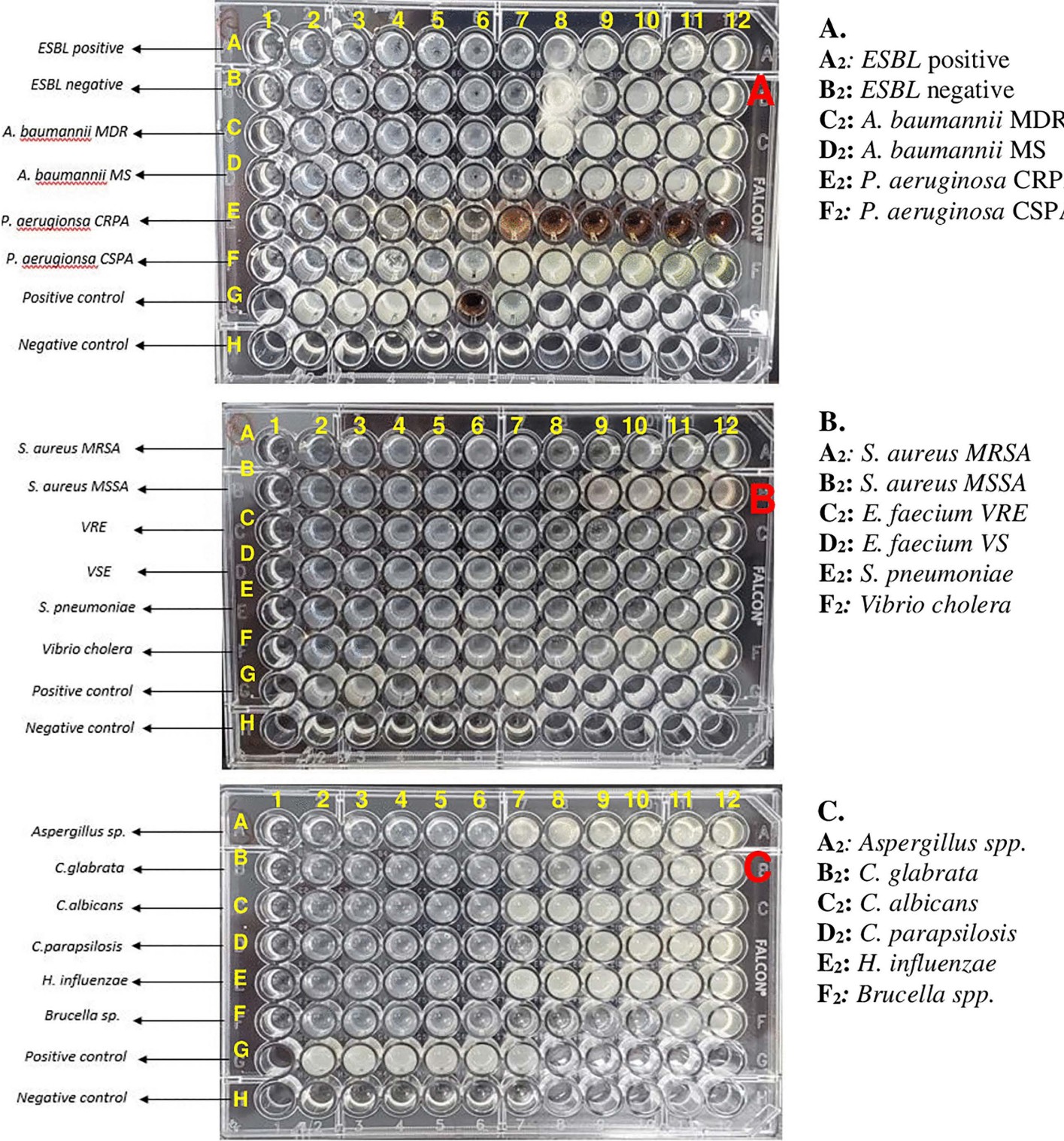

**A.**
**A₂**: *ESBL* positive
**B₂**: *ESBL* negative
**C₂**: *A. baumannii* MDR
**D₂**: *A. baumannii* MS
**E₂**: *P. aeruginosa* CRPA
**F₂**: *P. aeruginosa* CSPA

**B.**
**A₂**: *S. aureus MRSA*
**B₂**: *S. aureus MSSA*
**C₂**: *E. faecium VRE*
**D₂**: *E. faecium VS*
**E₂**: *S. pneumoniae*
**F₂**: *Vibrio cholera*

**C.**
**A₂**: *Aspergillus spp.*
**B₂**: *C. glabrata*
**C₂**: *C. albicans*
**D₂**: *C. parapsilosis*
**E₂**: *H. influenzae*
**F₂**: *Brucella spp.*

**Fig 3. Microtiter Plate Method.** Panels A to C illustrate the testing of various microorganisms, each arranged in a separate row and exposed to different concentrations of ZnO-NPs. The concentration range tested spanned from 9.7 to 310 µg/mL, achieved through serial two-fold dilutions. In each plate, row G contained all tested microorganisms without ZnO-NPs, serving as a positive growth control, while row H was filled with blank media to act as a

**Table 2. The mean of MIC values of the tested microorganisms.**

| No. | Microorganisms | The mean of the lowest dilution factor points where visible growth disappears (MIC$_{50}$) | The mean of MIC values indicated by the relative ZnO-NPs Conc. (µg/mL) |
|---|---|---|---|
| 1 | *ESBL +ve* | 256 | 39 |
| 2 | *ESBL -ve* | 512 | 19 |
| 3 | *A.baumannii MDR* | 64 | 150 |
| 4 | *A.baumannii MS* | 128 | 70 |
| 5 | *P.aeruginosa CRPA* | 32 | 310 |
| 6 | *P.aeruginosa CSPA* | 64 | 150 |
| 7 | *S.aureus MRSA* | 512 | 19 |
| 8 | *S.aureus MSSA* | 256 | 39 |
| 9 | *E.faecium VRE* | 256 | 39 |
| 10 | *E.faecium VSE* | 1024 | 9.7 |
| 11 | *S.pneumoniae* | 1024 | 9.7 |
| 12 | *Vibrio cholera* | 512 | 19 |
| 13 | *Aspergillus spp.* | 64 | 150 |
| 14 | *Candida glabrata* | 512 | 19 |
| 15 | *Candida albicans* | 64 | 150 |
| 16 | *Candida parapsilosis* | 128 | 70 |
| 17 | *H.influenzae* | 64 | 150 |
| 18 | *Brucella sp.* | 1024 | 9.7 |

widest zone generated around the ZnO-NPs disc in Gram – positive bacteria was 24.0 mm in diameter against MRSA isolates. While the narrowest zone of inhibition was 13.4 mm in diameter against VRE phenotype isolate. The clear zone surrounding the discs in "Fig 4" shows the antimicrobial action of synthesized ZnO-NPs, which prevents the growth of bacteria. The results showed that compared to Gram-negative bacteria, Gram-positive bacteria had larger inhibitory zones. Furthermore, ZnO-NPs effectively suppressed almost every pathogen type that was tested. The zone of inhibition formed by ZnO nanoparticles was found to be substantially comparable to the zone of inhibition produced by commercially available antibiotics, as shown in "Tables 3,4, S2, and S3". In order to eliminate the possibility that the observed impact was caused by the diluent solution used in the synthesis of ZnO-NPs (zinc acetate (ZnA)), a disc containing the relative concentration of zinc acetate alone was applied and examined. The results showed that the zinc acetate diluent had little effect on the zone of inhibition for ZnO-NPs. To evaluate whether the antimicrobial activity was solely attributed to the ZnO nanoparticles or partially due to the bioactive compounds present in *Phoenix dactylifera* root extract, comparative disc diffusion assays were performed using the crude extract alone. The results showed that the plant extract exhibited minimal or no inhibition zones against most bacterial strains tested. For example, the extract produced only 7.5 mm and 9.2 mm inhibition zones against *E. coli* and MRSA respectively, significantly lower than the 19.6 mm and 24.0 mm zones observed with ZnO-NPs. These findings confirm that the antimicrobial efficacy is significantly enhanced after nanoparticle synthesis

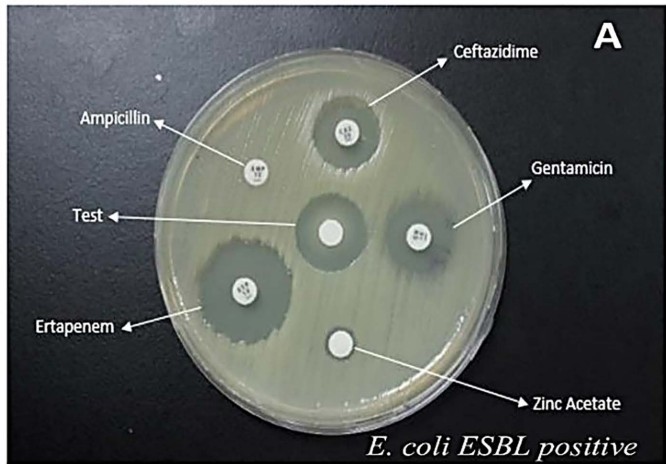

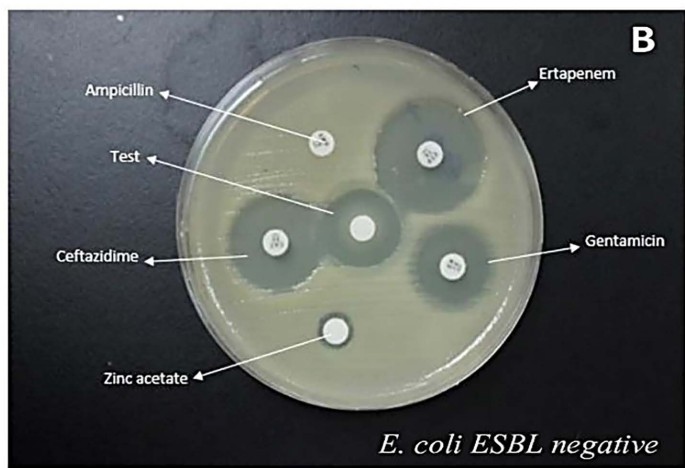

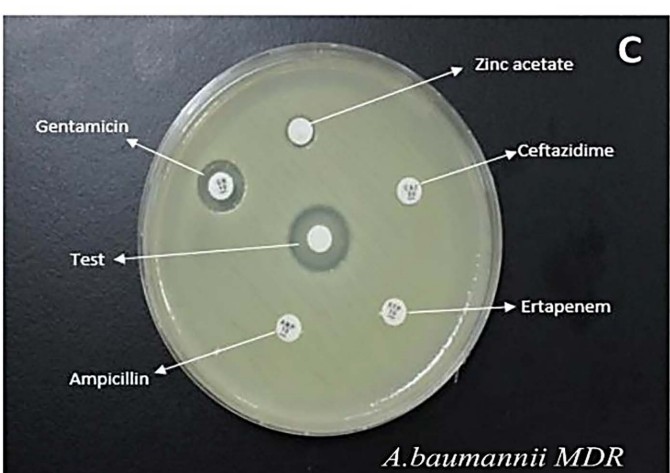

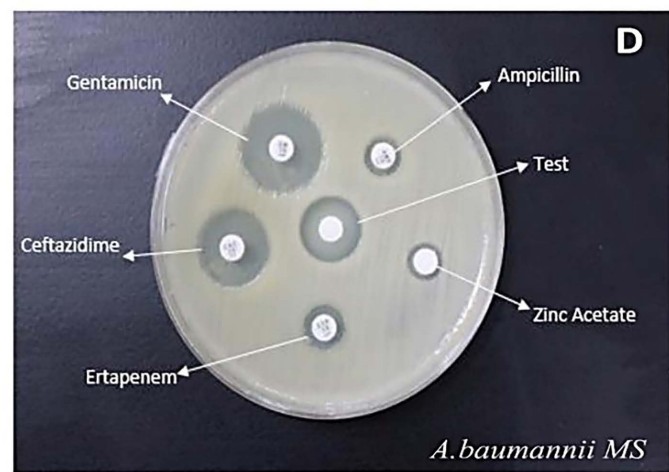

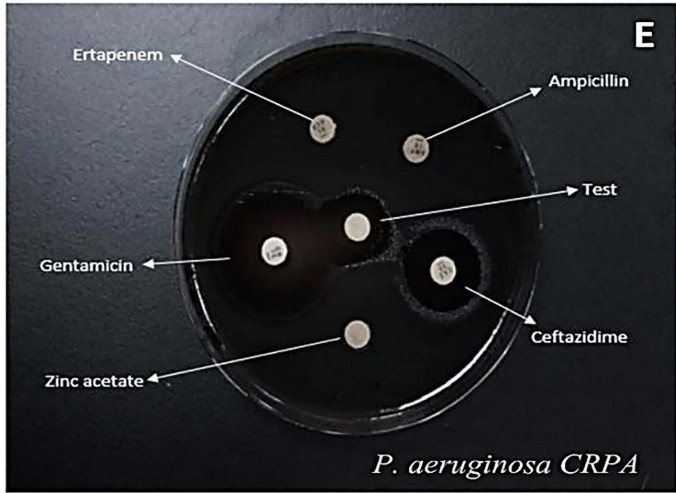

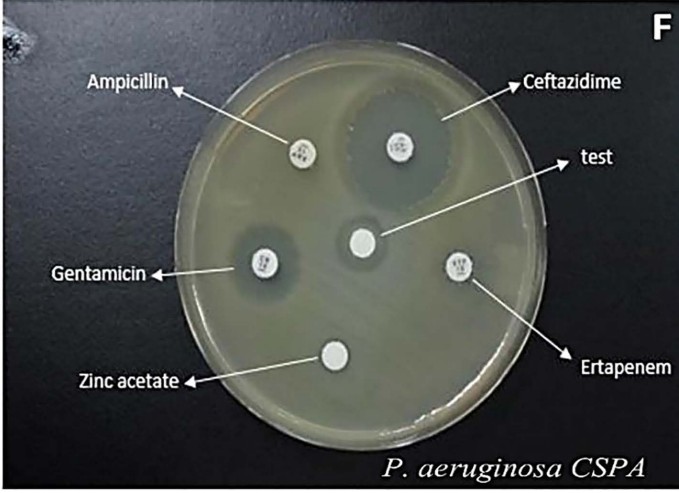

**Fig 4. Disc diffusion assay assessing the antimicrobial activity of ZnO-NPs against various Gram-positive and Gram-negative bacterial isolates.** Panels A to K display the antibacterial effects of ZnO-NPs alongside commercially available antibiotic discs for comparison. The central disc labeled "test" on each plate contains ZnO-NPs. Antibiotic discs were selected based on CLSI guidelines (CLSI, 2021) specific to each bacterial isolate. Additionally, a disc containing only zinc acetate was included on each plate as a diluent control. Abbreviations: Extended-spectrum beta-lactamases,

ESBLs; Multidrug-resistant bacteria, MDR; Carbapenem-resistant *Pseudomonas aeruginosa*, CRPA; Carbapenem- susceptible *Pseudomonas aeruginosa*, CSPA; *A. baumannii* Multisensitive, *A. baumannii* MS. Methicillin-resistant *Staphylococcus aureus*, MRSA; Methicillin-susceptible *Staphylococcus aureus*, MSSA; Vancomycin- susceptible *Enterococcus*, VSE; Vancomycin- resistant *Enterococcus*, VRE.

**Table 3. The mean of inhibition zone diameter for Gram-negative bacteria using disc diffusion assay.**

| Disc | ESBL positive | | ESBL negative | | *A. baumannii* MDR | | *A. baumannii* MS | | CPRA | | *P. aeruginosa* | |
|---|---|---|---|---|---|---|---|---|---|---|---|---|
| | Zone mm | I* | Zone mm | I | Zone mm | I | Zone mm | I | Zone mm | I | Zone mm | I |
| ZnO (Zinc Oxide NPs) | 19.4 | | 19.6 | | 16.0 | | 16.7 | | 17.1 | | 13.0 | |
| ZnA (Zinc Acetate) | No zone | | 9.0 | | No zone | | 9.0 | | No zone | | No zone | |
| Amp (Ampicillin) | No zone | R | No zone | R | No zone | R | 9.0 | R | No zone | R | No zone | R |
| CAZ (Ceftazidime) | 16.0 | R | 21.0 | S | No zone | R | 19.0 | I | 16.8 | I | 24.0 | S |
| ETP (Ertapenem) | 23.0 | S | 27.6 | S | No zone | R | 11.0 | R | No zone | R | 9.0 | R |
| CN (Gentamicin) | 13.4 | I | 16.7 | S | 11.6 | R | 20.0 | S | 19.0 | S | 16.8 | S |

Footnote: Zinc oxide, ZnO; Zinc acetate, ZnA; S, Susceptible; I, Intermediate; R, Resistant; Extended-spectrum beta-lactamases, ESBLs; Multidrug-resistant bacteria, MDR; Carbapenem-resistant *Pseudomonas aeruginosa*, CRPA; *A. baumannii* Multisensitive, *A. baumannii* MS.

**Table 4. The mean of inhibition zone diameter for Gram-positive bacteria using disc diffusion assay method.**

| Discs (Antibiotic) | MRSA | | MSSA | | VRE | | VSE | | *S. pneumoniae* | |
|---|---|---|---|---|---|---|---|---|---|---|
| ZnO (Zinc Oxide NPs) | 24.0 | | 17.0 | | 13.4 | | 14.1 | | 23.0 | |
| ZnA (Zinc Acetate) | 11.9 | | 10.4 | | No zone | | No zone | | 12.4 | |
| P (Penicillin) | No zone | R | 14.7 | R | No zone | R | No zone | R | 26.0 | S |
| OX (Oxacillin) | No zone | R | 20.0 | S | No zone | R | No zone | R | No zone | R |
| VA (Vancomycin) | 19.7 | S | 17.3 | S | No zone | R | 23.4 | S | 24.0 | S |
| LZD (Linezolid) | 35.0 | S | 28.0 | S | 24.7 | S | 27.5 | S | 30.0 | S |

Footnote: Zinc oxide, ZnO; Zinc acetate, ZnA; S, Susceptible; I, Intermediate; R, Resistant; Methicillin-resistant *Staphylococcus aureus*, MRSA; Methicillin-susceptible *Staphylococcus aureus*, MSSA; Vancomycin-susceptible *Enterococcus*, VSE.

and is not due to the plant extract alone. This comparative evaluation reinforces the functional role of ZnO-NPs as the primary antimicrobial agent in this study.

**The efficacy of ZnO-NPs against multidrug-resistant bacteria in water samples and various surface types**

In this study, the pour plate method was employed to assess the activity of ZnO-NPs against MDR bacteria in water samples and on various surfaces, including glass, benches, and stainless steel. The CFU/mL values for water samples, glass surfaces, and stainless-steel surfaces have been shown to reduce with incorporationofZnO-NPs "Fig 5". The reduction reached a zero CFU/mL after one hour of exposure to ZnO-NPs treatment. However, various reduction percentages relative to untreated control in CFU/mL formations were reported among water samples, glass surfaces, and stainless-steel surfaces at 5, 10, 15, 20, 25, 30, 60 mins, and overnight, as shown in "Tables 5, and S4". Furthermore, the findings of this study demonstrated that ZnO-NPs require 20 minutes of exposure on laboratory benches to completely inhibit the growth of both Gram-positive and Gram-negative bacteria, as shown in "Fig 6".

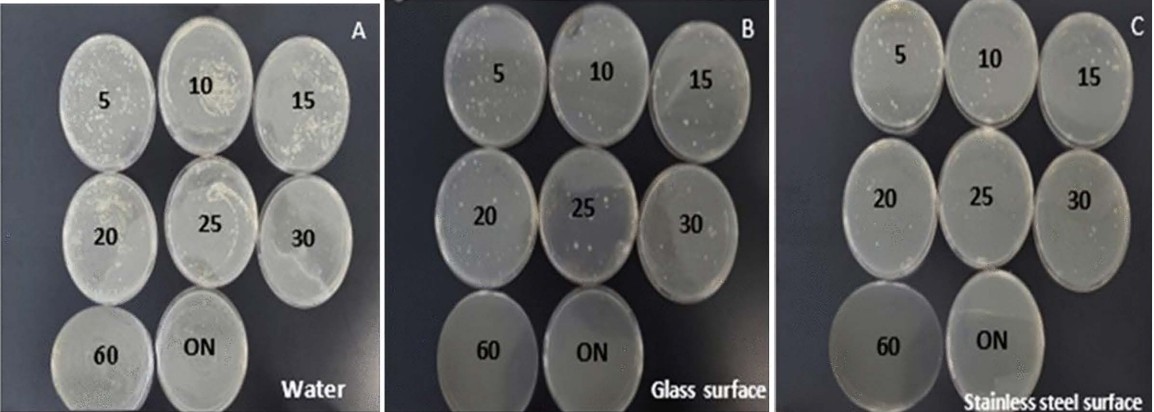

**Fig 5. Pour-plate method.** Each panel illustrates the disinfection effect of ZnO-NPs on different materials: water (Panel A), glass surface (Panel B), and stainless-steel surface (Panel C). The plates in each panel display microbial growth observed at various time intervals (5–60 minutes) and after overnight (ON) incubation.

**Table 5. Summary of CFU/mL and percent of reduction calculations on each sample type after exposure to ZnO-NPs at different time intervals, performed by pour plate assay.**

| Time of exposure | Water | | Glass | | Stainless steel | |
|---|---|---|---|---|---|---|
| | CFU/mL | % Of reduction | CFU/mL | % Of reduction | CFU/mL | % Of reduction |
| 0 min | 2650 | – | 950 | – | 520 | – |
| 5 mins | 2170 | 4.8 | 730 | 2.2 | 450 | 0.7 |
| 10 min | 1620 | 10.3 | 520 | 4.3 | 310 | 2.1 |
| 15 mins | 910 | 17.4 | 320 | 6.3 | 230 | 2.9 |
| 20 mins | 800 | 18.5 | 260 | 6.9 | 170 | 3.5 |
| 25 mins | 670 | 19.8 | 180 | 7.7 | 110 | 4.1 |

## The effect of ZnO-NPs on AFB isolates

The effect of ZnO-NPs on AFB isolates was investigated using CAP PT (CAP-PT E1-03 and E1-04 2021) with previously documented results. To investigate the impact of ZnO-NPs on AFB isolates and establish the bactericidal or bacteriostatic activity of ZnO-NPs, the time to positivity was determined. The results demonstrated that the ZnO-NPs were able to eliminate AFB bacteria. Therefore, the findings of this study showed that the tested ZnO-NPs are effective against a variety of clinically significant microorganisms, including AFB isolates "Fig 7".

## Discussion

This study's unique contribution lies in its use of a green synthesis approach, employing *Phoenix dactylifera* root extract, to biosynthesize ZnO nanoparticles, thereby offering a sustainable and eco-friendly alternative to conventional chemical methods. This study stands out further by conducting comprehensive antimicrobial evaluations against clinically relevant, multidrug-resistant bacteria and fungi, as well as practical disinfection assessments on water and environmental surfaces.

ZnO nanoparticles have been widely studied for their antimicrobial properties. Their effectiveness against various bacteria and fungi, including those resistant to conventional antibiotics, has been well documented. For instance, research has shown that ZnO-NPs exhibit significant antibacterial activity against *C. albicans*, *S. lutea*, *A. niger*, S.*pyogenes*, *P.*

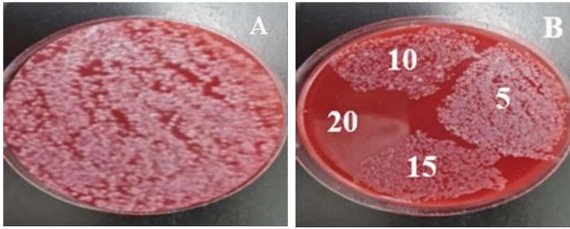

**Fig 6. The effect of ZnO-NPs as surface disinfectant on a contaminated laboratory bench.** A Benche area was divided into four equal sections (5×5 cm), each contaminated with saline containing bacteria and allowed to airdry. ZnO-NPs suspension was applied, and cultures were taken at 5, 10, 15, and 20 minutes using sterile swabs on blood agar. Plate A observed growth of lab bench surface without ZnO-NPs treatments (number of colonies was numerous). Plate B. Observe the growth after ZnO-NPs exposure at different time intervals, including 5, 10, 15, and 20 mins.

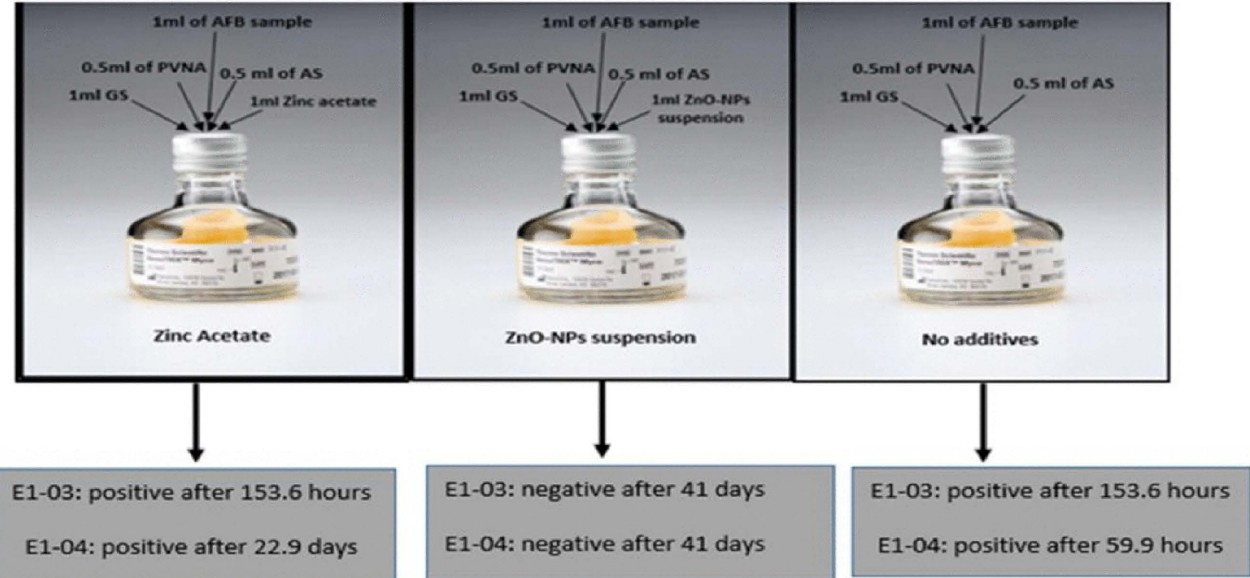

**Fig 7. Time-to-Positivity Assay.** Two CAP proficiency testing AFB strains (E1-03 and E1-04) were evaluated in three independent experiments. The instrument detects changes in internal bottle pressure and automatically flags bottles as positive for microbial growth. The mean time to positivity for each strain is indicated below the corresponding assay bottle.

*vulgaris*, *S. aureus*, *B. subtilis*, *P. aeruginosa*, *M. tuberculosis*, *B. megatherium*, *Escherichia coli*, and *K. pneumoniae*, including ESBL-producing strains [9–13].

The MIC results from this study show that a ZnO-NPs suspension at a concentration of 9.7 µg/mL significantly inhibits *E. faecium* VS, *S. pneumoniae*, and *Brucella spp*. In contrast, a much higher concentration of 310 µg/mL is required to inhibit the *P. aeruginosa* CRPA phenotype. The higher concentration required for the *P. aeruginosa* CRPA phenotype inhibition could be attributed to this pathogen's robust efflux system and biofilm formation abilities, which generally confer higher resistance to antimicrobials (2,28). These findings are significant as they demonstrate the varied susceptibility of different bacterial species to ZnO-NPs, with some bacteria being inhibited at much lower concentrations than others. On the other hand, the relatively low MIC for *E. faecium* VS, *S. pneumoniae*, and *Brucella spp*. suggests that these pathogens are more susceptible to the oxidative stress and membrane damage induced by ZnO-NPs [3,49–51]. Additionally, while the MIC concentration range in this study was selected based on prior literature and preliminary screening, we

acknowledge that expanding this range could help determine more precise and strain-specific inhibitory concentrations. Future studies may benefit from testing broader or finer concentration gradients to optimise nanoparticle efficacy across diverse microbial profiles.

To address the mechanism of interaction between ZnO-NPs and bacterial strains which was previously not detailed this study draws upon widely accepted antimicrobial mechanisms established in the literature. ZnO-NPs are known to act via multiple pathways: (i) the generation of reactive oxygen species (ROS), including hydrogen peroxide and superoxide radicals, which causes oxidative stress and damage DNA, proteins, and lipids within bacterial cells; (ii) the release of $Zn^{2+}$ ions that disrupt membrane integrity and interfere with enzymatic functions; and (iii) direct physical interaction with the bacterial membrane, leading to increased permeability, leakage of cellular contents, and eventual cell death. These mechanisms operate synergistically and explain the broad-spectrum activity of ZnO-NPs against both Gram-positive and Gram-negative organisms [4,52].

The effectiveness of ZnO-NPs against both Gram-positive and Gram-negative bacteria can be attributed to their ability to generate ROS, which cause oxidative stress, leading to cellular damage and disruption of membrane integrity. Studies have shown that metal-based nanoparticles, including ZnO-NPs, exhibit significant antimicrobial properties by inducing oxidative stress and generating ROS, which are detrimental to bacterial cells [50]. Additionally, the small size and high surface area of ZnO-NPs enhance their interaction with bacterial cells, further contributing to their antimicrobial efficacy [3]. These findings underscore the potential of ZnO-NPs as a versatile antimicrobial agent, particularly against pathogens with varying resistance mechanisms. Further research could optimize the use of ZnO-NPs in clinical settings, potentially in combination with other antibiotics to combat multidrug-resistant infections.

The disc diffusion method was employed to assess the antimicrobial activity of ZnO-NPs against various Gram-positive and Gram-negative bacteria. The study results revealed notable differences in inhibition zones between these bacterial groups. The widest inhibition zone observed for Gram-negative bacteria was 19.6 mm against *E. coli* and *K. pneumoniae* ESBL-negative phenotype isolates. Conversely, the narrowest zone measured was 13.0 mm against the *P. aeruginosa* CRPA phenotype isolate. In Gram-positive bacteria, the largest zone of inhibition, 24.0 mm, was noted against MRSA isolates, while the smallest, 13.4 mm, was against VRE phenotype isolates. These findings indicate that ZnO-NPs exhibit a broader range of inhibition against Gram-positive bacteria compared to Gram-negative ones, which is consistent with other studies [53–56], This could be due to the differences in cell wall structure, where Gram-positive bacteria have thicker peptidoglycan layers, which ZnO-NPs might more easily disrupt compared to the complex outer membrane of Gram-negative bacteria [53–55].

Comparative assays were conducted to investigate whether the antimicrobial activity observed in this study was solely due to the biosynthesized ZnO nanoparticles or partially attributable to bioactive phytochemicals in *Phoenix dactylifera* root extract. The crude plant extract alone exhibited minimal inhibition zones, ranging between 7.5 mm for *E. coli* and 9.2 mm for MRSA—markedly lower than the 19.6 mm and 24.0 mm observed with ZnO-NPs, respectively. This suggests that the extract exhibits weak or negligible antimicrobial properties and that the synthesis of ZnO-NPs significantly enhances its bioactivity. These results align with prior studies, such as those by Naser et al. (2021), which noted the limited antimicrobial action of *Phoenix dactylifera* or other plant extracts, but strong efficacy after nanoparticle synthesis [34].

Compared to other green-synthesized ZnO-NPs, such as those derived from *Equisetum diffusum* [Assad et al., 2023] and Wild Olive [Ullah et al., 2024], the present study results are consistent in showing enhanced antimicrobial activity only after nanoparticle formation [57,58]. This confirms that transforming plant phytochemicals into a nanoparticle matrix augments their efficacy by improving penetration, surface reactivity, and reactive oxygen species (ROS) generation. Thus, the ZnO-NPs synthesized using Phoenix dactylifera root extract act as the primary antimicrobial agent, with superior efficacy compared to the plant extract alone.

The antimicrobial mechanism of ZnO-NPs is attributed to multiple factors, including the release of Zn + 2 ions, which disrupt bacterial cell membranes, and the generation of ROS that induce oxidative stress, damaging cellular components

and leading to cell death [3,53,56,59,60]. These results highlight the potential of ZnO-NPs as an effective antimicrobial agent against various bacterial pathogens, with generally more potent activity against Gram-positive bacteria [55,61,62].

Furthermore, the antimicrobial activity of ZnO-NPs is comparable to that of commercially available antibiotics. This suggests that ZnO-NPs could be a viable alternative in antimicrobial applications, reinforcing the importance of continued research into their use against various pathogens [54]. This study also examines the impact of the diluent used in the synthesis of ZnO-NPs, finding that it does not significantly influence the observed antibacterial effects, which supports the conclusion that the ZnO-NPs themselves are responsible for the antimicrobial activity [59,63].

To contextualize the antimicrobial performance of biosynthesized ZnO nanoparticles (NPs), a comparative analysis was conducted with recent studies involving green-synthesized silver nanoparticles (AgNPs). Various plant-extract-functionalized AgNPs have demonstrated notable antibacterial activity across a spectrum of Gram-positive and Gram-negative pathogens, with differences attributed to nanoparticle size, surface charge, and capping agents. In comparison, green-synthesized ED-AgNPs prepared using *Equisetum diffusum* extract showed a maximum inhibition zone of 18 mm against *Listeria monocytogenes* and 10 mm against *E. coli*, with a reported average nanoparticle size of 63 nm (Jabbar et al., 2021). HEC-PA-capped AgNPs demonstrated enhanced antimicrobial properties, particularly against Gram-negative bacteria, supported by high colloidal stability (ZP = −35 mV) and a hydrodynamic size of 40 nm. However, inhibition zones were not quantified [64]. Furthermore, AgNPs synthesized using *Cotoneaster nummularia* extract exhibited antimicrobial activity and wound-healing potential, but did not report quantitative zone sizes [57].

Sulfonamide-functionalized AgNPs (AgNPs@PSBA) also displayed synergistic antibacterial activity with a nanoparticle size of 45 nm and good colloidal stability (ZP = −30 mV), although without inhibition zone data [65]. AgNPs derived from olive fruit extract exhibited exceptionally high antibacterial efficacy, particularly against E. coli, with an inhibition zone of 27 mm, attributed to strong capping by phytochemicals and a ZP of −40 mV [58]. Carboxylic acid-capped AgNPs (AgNPs@AA) synthesized using adipic acid also showed strong antimicrobial effects supported by a hydrodynamic size of 30 ± 5 nm and a zeta potential of −35.5 mV. However, specific inhibition zones were not provided [66].

In contrast to these studies, the present work offers a broader scope by integrating quantitative zone diameter analysis, MIC values, and disinfection efficacy on surfaces and water, as well as demonstrating activity against acid-fast bacilli (AFB) isolates. The synthesized ZnO-NPs were not only effective against a wide spectrum of Gram-positive and Gram-negative bacteria, including multidrug-resistant phenotypes, but also achieved complete CFU elimination from stainless steel, glass, and water samples within 60 minutes. Therefore, the present study provides a more comprehensive antimicrobial assessment than most comparable AgNP-based systems, thereby reinforcing the therapeutic and disinfectant potential of biosynthesized ZnO-NPs. The findings of the current study demonstrate the potent antibacterial properties of ZnO-NPs against MDR bacteria across various environments, including water samples and on surfaces like glass, stainless steel, and laboratory benches, where ZnO-NPs were found to effectively reduce CFU to zero after 1-hour exposure, with varying efficacy across different time intervals and surfaces. For example, the complete inhibition of bacterial growth on laboratory benches required only 20 minutes of exposure. Such a rapid reduction in bacterial colonies suggests that ZnO-NPs could be an effective disinfectant in various environments, potentially preventing the spread of infections in clinical settings and other areas prone to bacterial contamination. The specific mechanisms through which ZnO-NPs exert their antimicrobial effects include releasing metal ions, generating reactive oxygen species, and directly interacting with microbial cell membranes.

These mechanisms contribute to the disruption of bacterial cell function and integrity, leading to bacterial death [4,67,68]. Moreover, the study's findings are consistent with other research indicating that ZnO-NPs are effective against both Gram-positive and Gram-negative bacteria, enhancing their utility as a broad-spectrum antimicrobial agent [69,70]. Other studies have shown that these nanoparticles can be synergistically combined with antibiotics to enhance their bactericidal effects, which could be particularly valuable in combating antibiotic resistance [67,71]. Moreover, the structural properties of ZnO-NPs, play critical role in their antimicrobial efficacy [3,72]. This is particularly relevant for settings

where quick sterilization of surfaces is crucial to prevent the transmission of infectious agents. These results support ongoing research into the use of ZnO-NPs as alternative disinfectants and antimicrobial agents, especially in the context of increasing resistance to traditional antibiotics.

This study highlights the potential of ZnO-NPs as an effective surface disinfectant, particularly in their ability to inhibit microbial growth. The microtiter plate assay demonstrated that the formation ability of both bacteria and fungi is inversely proportional to the concentration of ZnO-NPs, indicating a clear dose-dependent relationship. Microbial communities are complex structures of microorganisms that adhere to surfaces and are protected by a self-produced matrix. The control of microbial propagation is crucial because these pathogens can confer resistance to antimicrobial agents and are involved in a variety of persistent bacterial infections [73]. The efficacy of ZnO-NPs in disrupting microbial structures, as shown by the decrease in formation with increasing concentrations of nanoparticles, aligns with other studies that have reported the anti-microbial properties of ZnO-NPs [74–77]. These studies suggest that ZnO-NPs can disrupt microbial integrity, enhance antibiotic penetration, or directly interact with microbial cells to inhibit their growth and viability. This finding supports the potential use of ZnO-NPs as a surface disinfectant, particularly in settings where microbial formation can lead to significant health and safety issues.

The antimicrobial properties of ZnO-NPs, including their ability to disrupt microbial membranes and induce oxidative stress, contribute to their effectiveness in microbial prevention [75,77,78]. This action is crucial for controlling infections, especially those caused by bacteria that are resistant to multiple drugs. Furthermore, the variability in microbial formation reduction across different microbial species highlights the broad-spectrum capabilities of ZnO-NPs. For instance, different studies have shown that ZnO-NPs can inhibit the formation in various bacterial strains, including *S. aureus* [76,79–83], *E. coli* [77,84], *P. aeruginosa* [84], and *K. pneumonia* [74,75], underlining the versatility of ZnO-NPs in antimicrobial applications. As noted in this study, the specific activity of ZnO-NPs against different bacterial phenotypes aligns with the broader research landscape. Moreover, the significance of ZnO-NPs extends to their combination with other agents to synergize effects and enhance antimicrobial and antivirulence activities against resistant strains. For instance, studies have shown that ZnO-NPs can work in conjunction with established antibiotics or bioactive compounds like piperine, enhancing their effectiveness and potentially reducing the antibiotics needed, which can help in the fight against antibiotic resistance [78,84]. Overall, the consistent reduction in microbial formation across various studies validates the potential of ZnO-NPs as an effective antimicrobial agent, which could be further explored in healthcare and industrial applications to manage and prevent infections.

The differential responses to ZnO-NPs, especially between microorganisms exhibiting MDR and multisensitive behaviours, are particularly insightful. MDR organisms require higher concentrations of ZnO-NPs to effectively inhibit their microbial formation, which aligns with their general resistance traits against antimicrobial agents. This resistance could be due to various adaptive mechanisms these bacteria have developed to survive in hostile environments. In contrast, multisensitive microbes, which are generally more susceptible to antibiotics, also show lesser resistance to the inhibitory effects of ZnO-NPs at lower concentrations [76,82,85]. These insights highlight the complex interactions between microbial characteristics and the antimicrobial actions of nanoparticles, such as ZnO-NPs. They also emphasize the need for tailored approaches when using nanoparticles as antimicrobial agents, particularly in environments where related infections are prevalent. In addition, these findings are important for developing targeted strategies using ZnO-NPs in combating associated infections, particularly in clinical settings where these communities can significantly contribute to the persistence and severity of infections. Further research could explore the specific interactions and mechanisms at play, potentially leading to more effective antimicrobial treatments that can be tailored according to the resistance profiles of the pathogens involved.

Furthermore, the study investigated the effect of ZnO-NPs on acid-fast bacilli (AFB) isolates using the CAP PT E1-03 and E1-04 2021. The results showed that ZnO-NPs (0.01 g/mL) effectively eliminated AFB bacteria, demonstrating their bactericidal properties against these clinically significant microorganisms. This is particularly notable because AFB, which

includes mycobacteria like *Mycobacterium tuberculosis*, is known for its resistance to many conventional antibiotics [86]. According to a prior study, 90% of *Mycobacterium spp*. bacterial colonies destroyed after being exposed to 500–1000 µg/mL of ZnO-NPs for just 6 hours [12]. A substantial decrease in the development of drug-resistant *S. aureus*, *Mycobacterium smegmatis*, and *Mycobacterium bovis* was reported when they were treated with a combination of zinc oxide nanoparticles and a low dosage of the anti-tuberculosis treatment, rifampicin (0.7 µg/mL) [12].

These pathogenic bacteria were effectively eradicated after being cultured for 24 hours with a concentration of 1000 µg/mL of zinc oxide nanoparticles. Thus, it may be speculated that administering the exact dosage repeatedly can lead to a complete recovery in patients with these infectious diseases [12]. However, a previous study discovered that the minimum inhibitory concentration of rifampicin was reduced by fourfold when ZnO-NPs were combined with rifampicin against wild-type (WT) *Mycobacteria smegmatis*, resulting in a subinhibitory concentration of only 32 µg/mL [87]. On the other hand, a previous study found that at 12.5 µg/mL, ZnO-NPs ranging in size from 12 nm to 53 nm inhibited the development of *Mycobacteria tuberculosis* [88]. The ability of ZnO-NPs to target and eliminate AFB suggests that these nanoparticles could be a valuable tool in treating infections caused by such infectious bacteria. Additionally, the effectiveness of ZnO-NPs against a range of pathogens, including drug-resistant strains, highlights their potential as a versatile antimicrobial agent in clinical settings. Overall, this study supports the potential use of ZnO-NPs in treating infections caused by AFB and other clinically significant microorganisms, offering a promising alternative to traditional antibiotics, especially in the context of rising antimicrobial resistance.

Given their potent antimicrobial activity, especially against multidrug-resistant bacteria, ZnO-NPs synthesized in this study may hold practical value in real-world applications. These include their potential incorporation into hand sanitizers, surface disinfectants, wound dressings, and antimicrobial coatings for medical devices. Such applications are supported by the observed ability of ZnO-NPs to completely inhibit bacterial growth on various surfaces within short exposure times. However, to enable clinical or commercial use, future studies should investigate suitable formulation strategies, stability, and safety assessments for both human and environmental health.

Finally, in addition to nanoparticles, various interdisciplinary strategies such as smart materials, bioengineered devices, and advanced diagnostics have emerged to combat antimicrobial resistance. Innovations include self-assembling biomaterials, soft robots for targeted therapy, aggregation-induced emission-based tools for real-time pathogen detection, and systems for multi-pathogen diagnostics [89–91]. Biological factors like microbiota and herbal compounds also influence drug behaviour [92,93]. Nanoparticles can be effectively integrated into these technologies to enhance targeted antimicrobial action.

## Conclusion

The ZnO-NPs exhibit broad-spectrum antimicrobial properties, effective against both Gram-positive and Gram-negative bacteria, including MDR strains. Their ability to rapidly reduce microbial load on surfaces positions them as promising agents for clinical and environmental applications. Further research into optimizing their use, particularly in combination with other antimicrobials, could enhance their efficacy against infections that are resistant to treatment.

In this study, ZnO-NPs synthesized using *Phoenix dactylifera* root extract demonstrated minimum inhibitory concentrations (MICs) as low as 9.7 µg/mL for *E. faecium*, *S. pneumoniae*, and *Brucella spp*., while higher concentrations (310 µg/mL) inhibited resistant strains like *P. aeruginosa* (CRPA). Disc diffusion assays revealed strong antibacterial zones, ranging from 24.0 mm against MRSA to 19.6 mm against ESBL-negative *E. coli* and *K. pneumoniae*. ZnO-NPs also eliminated CFUs on stainless steel, glass, and water surfaces within 60 minutes and inhibited AFB isolates, highlighting rapid bactericidal potential across clinical and environmental settings.

Future aspects of this research work include the development of ZnO-NP-based formulations such as topical creams, wound dressings, and hydrogel coatings for infection control in hospitals and outpatient settings. Their integration with conventional antibiotics should also be explored to evaluate possible synergistic effects, potentially reducing required

drug dosages and mitigating resistance. Additionally, ZnO-NPs may be employed as antimicrobial coatings on medical devices, surgical tools, and hospital surfaces to minimize healthcare-associated infections (HAIs). Their application in air and water purification systems, such as antimicrobial filters or sprays, holds promise for improving hygiene in both clinical and resource-limited environments. To enable safe clinical translation, comprehensive in various studies and cytotoxicity assessments are necessary to establish biocompatibility, particularly for skin, pulmonary, and systemic exposure. Mechanistic investigations are also warranted to elucidate nanoparticle-bacteria interactions and assess the risk of resistance development upon prolonged exposure. Ultimately, addressing the scalability of green synthesis methods and assessing the environmental impact of ZnO-NP disposal will be crucial for the sustainable and responsible use of these materials in healthcare and industry.

## Supporting information

**S1 Table. The mean of MIC values of the tested microorganisms.**
(DOCX)

**S2 Table. The mean of inhibition zone diameter for Gram negative bacteria using disc diffusion assay.**
(DOCX)

**S3 Table. The mean of inhibition zone diameter for Gram positive bacteria using disc diffusion assay.**
(DOCX)

**S4 Table. Percent of reduction in CFU/mL calculations on each sample type after exposure to ZnO-NPs at different time intervals performed by pour plate assay.**
(DOCX)

## Author contributions

**Formal analysis:** Dalia Alqaffa, Ali M. Atoom, Mai Abdel Haleem A. Abusalah.

**Investigation:** Dalia Alqaffa, Ali M. Atoom.

**Methodology:** Dalia Alqaffa, Ali M. Atoom.

**Resources:** Rana Abu Huwaij, Awatef Al-Kaabneh, Maher A. Sughayer.

**Supervision:** Ali M. Atoom.

**Visualization:** Dalia Alqaffa, Bayan Tayseer Alzubi.

**Writing – original draft:** Dalia Alqaffa, Ali M. Atoom, Mai Abdel Haleem A. Abusalah, Manal Abdel Haleem A. Abusalah.

**Writing – review & editing:** Dalia Alqaffa, Ali M. Atoom, Mai Abdel Haleem A. Abusalah, Bayan Tayseer Alzubi, Rana Abu Huwaij, Awatef Al-Kaabneh, Manal Abdel Haleem A. Abusalah, Maher A. Sughayer.

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
