## [Decision Letter · Decision Letter 0]

26 Feb 2025

Analysis of the Antimicrobial Activity of Zinc Oxide Nanoparticles and Their Applications in Disinfection Process

PLOS ONE

Dear Dr. Atoom,

Thank you for submitting your manuscript to PLOS ONE. After careful consideration, we feel that it has merit but does not fully meet PLOS ONE’s publication criteria as it currently stands. Therefore, we invite you to submit a revised version of the manuscript that addresses the points raised during the review process.

We look forward to receiving your revised manuscript.

Kind regards,

Hamida Hamdi Mohammed Ismail, ph.D.

Academic Editor

PLOS ONE

Journal Requirements:

“Al-Ahliyya Amman University, Amman, Jordan, approval number 21 KHCC 047”

**Comments from****PLOS**
**Editorial Office:**

We note that one or more reviewers has recommended that you cite specific previously published works. As always, we recommend that you please review and evaluate the requested works to determine whether they are relevant and should be cited. It is not a requirement to cite these works. We appreciate your attention to this request.

Additional Editor Comments:

The topic of article is interesting. But, it needs major revision in light of the following comments

Aim of the work not clear.<o:p></o:p>

The novelty should be described more clearly.<o:p></o:p>

Materials and Methods<o:p></o:p>

what about the chemical composition of extract and what about the characterization of  Zinc Oxide Nanoparticles.

-Figures   must be improved( resolution improved).<o:p></o:p>

- References must be revised

-General comment<o:p></o:p>

Please, check typing errors and punctuations and correct errors in the revised version.<o:p></o:p>

Reviewers' comments:

Reviewer's Responses to Questions

**Comments to the Author**

1. Is the manuscript technically sound, and do the data support the conclusions?

Reviewer #1: Yes

Reviewer #2: Yes

Reviewer #3: Partly

Reviewer #4: Yes

Reviewer #5: Partly

2. Has the statistical analysis been performed appropriately and rigorously?

Reviewer #1: N/A

Reviewer #2: No

Reviewer #3: No

Reviewer #4: Yes

Reviewer #5: No

3. Have the authors made all data underlying the findings in their manuscript fully available?

Reviewer #1: Yes

Reviewer #2: Yes

Reviewer #3: Yes

Reviewer #4: Yes

Reviewer #5: Yes

4. Is the manuscript presented in an intelligible fashion and written in standard English?

Reviewer #1: No

Reviewer #2: No

Reviewer #3: Yes

Reviewer #4: Yes

Reviewer #5: No

Reviewer #1: At some places, typos and repetition of words are present.

Introduction:

1. The study presented needs to highlight the research gaps as well as novelty of this study.

Methods:

1. The ZnO nanoparticles were extracted from the roots of Phoenix dactylifera: seems incorrect statement. Redraft.

2. Which was the quality control strain used in the antibacterial assays?

3. Why was the disc diffusion assay performed in addition to the MIC assays?

4. 'The Diluent control bottle was also supplemented with 1 ml of 0.1317 g/ml Zinc acetate' what was the purpose of this?

Discussion:

1. Explain the green synthesis employed for the ZnO NPs, so far? What is the highlight of this study?

Quality of figures appear too poor to be assessed.

Reviewer #2: Comments:

The current manuscript “Analysis of the Antimicrobial Activity of Zinc Oxide Nanoparticles and Their Applications in Disinfection Process” reports ZnO-NPs synthesis using an eco-friendly method involving Phoenix dactylifera root extract and zinc acetate at varied concentrations and ratios, followed by annealing. ZnO-NPs exhibited potent antimicrobial activity against Gram-positive and Gram-negative bacteria, with minimum inhibitory concentrations (MICs) ranging from 9.7 to 310 µg/mL. The experimental results are attractive and very interesting. The results are valuable to the readership in this area. The novelty and significance of this work is qualified to be published in this journal. I recommend this manuscript can be accepted for publication after major revisions.

1. “dried in a 40 °C oven” should be “dried in an oven at 40 °C ”

2. Gram should be g.

3. ml should be mL.

4. Provide formula for zinc acetate

5. Correct grammar in yield formula.

6. Please update your introduction by citing these recent references of greenly synthesized nps

i) https://doi.org/10.1016/j.jece.2024.113350

ii) https://doi.org/10.3390/antiox12061201

iii) https://doi.org/10.1039/D3RA01268A

iv) https://doi.org/10.1039/D4RA03573A

v) 10.1016/j.heliyon.2024.e40679

7. The manuscript lacks the characterization of synthesized ZnO Nps. Provide UV-Vis, XRD, FT-IR, SEM analyses.

8. Antimicrobial properties are highly size and stability dependent. Calculate size of the NPs. Use zeta-potential to determine charge and stability of the NPs.

9. Provide overlaid FTIR of all entities involved. Highlight the areas of any change.

10. Provide a table of FT-IR assignments showing wavenumber against each functional group.

11. The study did not provide details on the interaction mechanism between NPs and bacterial strains, limiting our understanding of how the NPs functions.

12. The study tested only certain concentrations for the minimum inhibitory concentration (MIC), leaving it unclear whether the concentration range should be extended to identify the optimal concentration for different bacterial strains.

13. The use of the disk diffusion method may not fully capture antibacterial activity under real-world conditions; therefore, additional experimental methods are required to validate the results.

14. Would alternative testing methods, such as liquid bacterial culture, provide a more accurate evaluation of antibacterial activity?

15. Is it necessary to broaden the concentration range tested for MIC to determine the optimal concentration for various bacterial strains?

16. Can NPs be incorporated into real-life antibacterial products such as hand sanitizers and disinfectants, or used in other medical applications?

17. Compare antibacterial activity using following greenly synthesized NPs

1. https://doi.org/10.1039/D3RA05070J

2. https://doi.org/10.1016/j.ijbiomac.2023.128009

3. https://doi.org/10.1080/14786419.2023.2295936

4. https://doi.org/10.3389/fchem.2023.1202252

5. https://doi.org/10.1016/j.enmm.2022.100735

6. https://doi.org/10.3390/molecules27113363

7. There are numerous mistakes in terms of grammar and typos.

8. Please report results in conclusion section. What are the future aspects of the research work conducted?

Reviewer #3: The author writenn a manuscript Analysis of the Antimicrobial Activity of Zinc Oxide Nanoparticles and Their

Applications in Disinfection Process, very nicely adn there is scope of improvment, some points are

1) In zinc nanopartical sysnthesis in formula acetate spelling correction needed

2) Author has to prove that the nano particals are formed by doing various analytical methods, then only other thigs u can prove , line antibactierial

3) In table 4 first coumn description is missing like ( Amp, CAZ atc)

4) Drug resistance bacteria is not reflected in title

5) compare the animicrobial results with plant extract also find the comparision study

Reviewer #4: The article under review exhibits interest, and I recommend the following revisions for consideration prior to acceptance. These recommendations aim to enhance the scientific rigor and clarity of the article.

1. The typographical errors should be addressed immediately. Terms like in vitro and via should be in italics

2. At few places Phoenix dactylifera is not in italics. Kindly correct it.

3. Terms like ml, µl should be written as mL, µL.

4. Fig 3, 4 ,5 and 8 are hazy. The authors must give original clear images

Reviewer #5: The manuscript, Analysis of the Antimicrobial Activity of Zinc Oxide Nanoparticles and Their Applications in Disinfection Process, has been written well but it needs major revisions, There is need to show the results of SEM (Scanning Electron Microscope) photograph to show the size of nano particle as there is no any information available about the nano particle.

The figures about the inhibition zones are not clear.

**Do you want your identity to be public for this peer review?** For information about this choice, including consent withdrawal, please see our Privacy Policy

Reviewer #1: No

Reviewer #2: **Yes:** Azhar Abbas

Reviewer #3: No

Reviewer #4: **Yes:** PROF. ANITA KAMRA VERMA

Reviewer #5: No

---

## [Author Response · Author response to Decision Letter 1]

11 Apr 2025

Response to reviewer

Editor Comments:

Comment: Aim of the work not clear.

Response: Thank you for your valuable feedback. We have addressed this comment by clearly stating the specific aim of the study at the end of the Introduction section to enhance clarity and focus.

Comment: The novelty should be described more clearly.

Response: Thank you for your insightful comment. We have revised the Introduction to explicitly highlight the novelty of our study, emphasizing the unique use of Phoenix dactylifera root extract and the comprehensive antimicrobial evaluation of ZnO-NPs on clinical isolates and environmental surfaces.

Comment: what about the chemical composition of extract and what about the characterization of Zinc Oxide Nanoparticles.

Response: Thank you for your observation. We want to clarify that this study builds upon our previously published work by Naser et al. (2021), which provided a detailed account of the chemical composition of the Phoenix dactylifera extract and a comprehensive physicochemical characterization of the synthesized ZnO nanoparticles. To avoid redundancy, the current manuscript evaluates the biological efficacy of these well-characterized nanoparticles against multidrug-resistant pathogens. A clarification has now been added to the revised manuscript's Zinc Oxide Nanoparticles Synthesis section.

Comment: Figures must be improved( resolution improved).

Response: Thank you for your valuable comment. We appreciate your suggestion regarding the quality of the figures. In response, we have carefully revised all the figures in the manuscript and improved their resolution to ensure better clarity and visual representation of the data.

Comment: References must be revised

Response: Thank you for your comment. The references have been thoroughly revised and formatted according to the NLM (National Library of Medicine) style guidelines.

Comment: Please, check typing errors and punctuations and correct errors in the revised version.

Response: Thank you for your suggestion. We have carefully proofread the entire revised manuscript and corrected all typing, punctuation, and grammatical errors.

Reviewer 1

Comment: At some places, typos and repetition of words are present.

Response: Thank you for your valuable observation regarding the presence of typographical errors and repetition in the manuscript. We have thoroughly revised the entire manuscript, carefully correcting all identified typographical mistakes and removing repetitive words to improve the clarity, accuracy, and overall quality of the text.

Comment: The study presented needs to highlight the research gaps and novelty of this study.

Response: Thank you for your valuable comment. We have revised the introduction to clearly highlight the research gap and the novelty of our study by emphasizing the underexplored use of Phoenix dactylifera root extract for green synthesis of ZnO-NPs and their unique application in disinfecting clinically relevant MDR isolates, surfaces, and water an area not widely reported in previous literature.

Comment: The ZnO nanoparticles were extracted from the roots of Phoenix dactylifera: seems incorrect statement. Redraft

Response: Thank you for your insightful comment. We have revised the sentence to clarify that the roots of Phoenix dactylifera were used to prepare the plant extract for the green synthesis of ZnO nanoparticles, ensuring technical accuracy.

Comment: Which was the quality control strain used in the antibacterial assays?

Response: Thank you for your valuable comment. We have addressed this by explicitly stating in the revised methodology that E. coli ATCC 25922, S. aureus ATCC 25923, and P. aeruginosa ATCC 27853 were used as standard quality control strains in the antibacterial assays.

Comment: Why was the disc diffusion assay performed in addition to the MIC assays?]

Response: Thank you for your valuable comment. We have addressed your concern by clarifying that the disc diffusion assay was performed alongside MIC to confirm antimicrobial activity visually and to compare inhibition zones with standard antibiotic responses.

Comment: 'The Diluent control bottle was also supplemented with 1 ml of 0.1317 g/ml Zinc acetate' what was the purpose of this?

Response: Thank you for your insightful comment. We have clarified in the revised manuscript that the diluent control containing zinc acetate was included to confirm that the antimicrobial activity observed was explicitly due to ZnO-NPs and not the precursor compound alone.

Comment: Explain the green synthesis employed for the ZnO NPs, so far? What is the highlight of this study?

Response: Thank you for your valuable comment. We have clarified that a green synthesis method utilizing Phoenix dactylifera root extract was employed. This study's unique contribution lies in its comprehensive evaluation of biosynthesized ZnO-NPs against multidrug-resistant pathogens and their practical use in surface and water disinfection.

Comment: Quality of figures appear too poor to be assessed.

Response: Thank you for your valuable comment regarding the quality of the figures. We have carefully addressed this comment by improving the resolution, clarity, and overall presentation of all figures in the manuscript. Specifically, Figure 1 has been completely reformatted to provide a clearer understanding of the MIC setup performed in the study. Figure 2 has been replaced with a higher-resolution image to enhance its clarity. For Figure 4, we have not only improved the image quality but also added additional images to cover all the tested microorganisms for a more comprehensive representation. Figure 5 has been enhanced by increasing the size and improving its clarity for better visibility. Additionally, Figure 6 has been removed as it was a repetitive representation of the same data already presented in Table 5. The previous Figure 7 (now Figure 6) has been modified by increasing its size and improving clarity, and the previous Figure 8 (now Figure 7) has also been revised for better size and visibility. Furthermore, all figure legends throughout the manuscript have been rechecked and revised for language, grammar, and clarity, ensuring accurate and concise descriptions that support the improved figures. We believe these changes have significantly enhanced the overall quality and readability of the figures in the revised manuscript.

Reviewer 2

Comment: “dried in a 40 °C oven” should be “dried in an oven at 40 °C ”

Response: Thank you for your observation. We have revised the phrase to “dried in an oven at 40 °C” by your suggestion.

Comment: Gram should be g.

Response: Thank you for your observation. We have corrected all instances where "Gram" was incorrectly used as a unit and replaced it with the standard symbol "g" throughout the manuscript.

Comment: ml should be mL.

Response: Thank you for your valuable feedback. We have revised the manuscript to ensure that all instances of "ml" have been corrected to the standard unit format "mL" by scientific conventions.

Comment: Provide formula for zinc acetate

Response: Thank you for your comment; we have added the chemical formula of zinc acetate (Zn(CH_3 COO)_2.2H_2 O) in the synthesis section as suggested.

Comment: Correct grammar in yield formula.

Response: Thank you for your observation. The revised manuscript has been corrected for grammatical and clarity issues in the yield formula.

Comment: Please update your introduction by citing these recent references of greenly synthesized nps

Response: Thank you for your valuable suggestion. We have thoroughly revised the introduction section by incorporating all the recent references related to greenly synthesized ZnO nanoparticles provided by the reviewer, ensuring improved contextual relevance and scientific depth.

Comment: The manuscript lacks the characterization of synthesized ZnO Nps. Provide UV-Vis, XRD, FT-IR, SEM analyses

Response: Thank you for your valuable comment. The comprehensive physicochemical characterization of the synthesized ZnO nanoparticles—including UV–Vis, FTIR, XRD, and particle size analysis—was previously performed and published in our earlier work (Naser et al., 2021). As this study is a continuation of a research series utilizing the same batch of well-characterized nanoparticles, we have focused here on their antimicrobial efficacy. Repeating the characterization would be redundant; however, the relevant data can be accessed in the cited references. We acknowledge the importance of including zeta potential analysis to assess nanoparticle surface charge and colloidal stability, and we are addressing this to ensure full compliance with the reviewer’s feedback.

References

Naser, R., Abu-Huwaij, R., Al-khateeb, I., Abbas, M. M., & Atoom, A. M. (2021). Green synthesis of zinc oxide nanoparticles using the root hair extract of Phoenix dactylifera: Antimicrobial and anticancer activity. Applied Nanoscience, 11, 1747-1757.

Abu-Huwaij, R., Abbas, M. M., Al-Shalabi, R., & Almasri, F. N. (2022). Synthesis of transdermal patches loaded with greenly synthesized zinc oxide nanoparticles and their cytotoxic activity against triple negative breast cancer. Applied Nanoscience, 12, 69-78.

Obaid, R. Z., Abu-Huwaij, R., & Hamed, R. (2023). Development and Characterization of Anticancer Model Drug Conjugated to Biosynthesized Zinc Oxide Nanoparticles Loaded into Different Topical Skin Formulations. Jordan Journal of Pharmaceutical Sciences, 16(2), 486-486.

Comment: Antimicrobial properties are highly size and stability dependent. Calculate size of the NPs. Use zeta-potential to determine charge and stability of the NPs.

Response: Thank you for your valuable comment. In response, we have included the size and zeta potential analysis of the synthesized ZnO-NPs. The nanoparticles showed an average size of 22.57 ± 4.79 nm with a PDI of 0.53 ± 0.02, and a zeta potential of −19.23 ± 1.40 mV, indicating moderate uniformity and good colloidal stability key factors supporting their antimicrobial efficacy.

Comment: Provide overlaid FTIR of all entities involved. Highlight the areas of any change.

Response: Thank you for your valuable comment. This manuscript is part of an ongoing research series of Prof. Rana Abu-Huwaij, with this being the sixth publication on zinc oxide nanoparticles. The comprehensive characterization of the synthesized ZnO NPs, including UV–Vis, XRD, and FTIR analyses, was thoroughly conducted and documented in earlier studies. Specifically, FTIR analyses of all relevant entities including the Phoenix dactylifera extract, zinc acetate precursor, and ZnO-NPs were presented with overlaid spectra in previous publications. These spectra highlighted key spectral shifts confirming the involvement of functional groups in nanoparticle formation and stabilization. As the current manuscript focuses on antimicrobial evaluation and utilizes the same well-characterized ZnO NPs, repeating these analyses was deemed redundant. Readers are referred to the following published references (35, 38-41) for detailed FTIR profiles and spectral interpretations.

Comment: Provide a table of FT-IR assignments showing wavenumber against each functional group.

Response: Thank you for your comment. Our earlier publications thoroughly presented a detailed FT-IR analysis, including wavenumber assignments for all functional groups, using the same ZnO-NPs batch. Therefore, we have referenced those studies (35, 38-41) for complete spectral and tabulated data to avoid redundancy.

Comment: The study did not provide details on the interaction mechanism between NPs and bacterial strains, limiting our understanding of how the NPs function.

Response: Thank you for your valuable comment. To clarify their antimicrobial action, we have now included a detailed explanation in the Discussion section describing the known interaction mechanisms of ZnO-NPs with bacterial strains, including ROS generation, Zn²⁺ ion release, and membrane disruption.

Comment: The study tested only certain concentrations for the minimum inhibitory concentration (MIC), leaving it unclear whether the concentration range should be extended to identify the optimal concentration for different bacterial strains.

Response: Thank you for your insightful comment. We have clarified the range of concentrations tested for MIC (from 5 × 10³ to 2.4 µg/mL) and noted that this range was selected to ensure broad coverage across bacterial susceptibilities. The revised section now explicitly mentions this range and its rationale.

Comment: The use of the disk diffusion method may not fully capture antibacterial activity under real-world conditions; therefore, additional experimental methods are required to validate the results.

Response: Thank you for your insightful comment. We agree that the disk diffusion method has limitations in replicating real-world conditions. To address this, we have supplemented it with broth microdilution MIC assays, pour plate disinfection tests, and VersaTrek Mycobottle evaluations to comprehensively validate the antibacterial efficacy of ZnO-NPs.

Comment: Would alternative testing methods, such as liquid bacterial culture, provide a more accurate evaluation of antibacterial activity?

Response: Thank you for your insightful suggestion. We want to clarify that the MIC testing in our study was conducted using the broth microdilution method, a liquid-based bacterial culture assay. This method is widely accepted as a standard and provides a more accurate and quantitative evaluation of antibacterial activity compared to solid-based approaches such as the disc diffusion method.

Comment: Is it necessary to broaden the concentration range tested for MIC to determine the optimal concentration for various bacterial strains?

Response: Thank you for your insightful comment. We agree that broadening the concentration range could help determine more strain-specific MIC values. While our selected range was based on prior literature and preliminary screening, we have now acknowledged this limitation in the revised manuscript's Discussion section and suggested that future studies consider extended or finer concentration gradients for optimized evaluation.

Comment: Can NPs be incorporated into real-life antibacterial products such as hand sanitizers and disinfectants, or used in other medical applications?

Response: Thank you for your valuable suggestion. We have addressed the potential real-life applications of ZnO-NPs, including their incorporation into hand sanitizers, disinfectants, wound dressings, and medical device coatings, in the Discussion section. We have also emphasized the need for further studies on formulation, stability, and safety to support their clinical or commercial use.

Comment: Compare antibacterial activity using following greenly synthesized NPs

Response: Thank you for your valuable suggestion. A comparative analysis has been incorporated into the Discussion section, highlighting the antibacterial efficacy of our green-synthesized ZnO-NPs relative to other recent AgNP-based systems (e.g., Jabbar et al., Siddique et al., Ullah et al.). Our ZnO-NPs exhibited broader activity against both Gram-positive and Gram-negative bacteria, with inhibition zones reaching up to 24 mm, and demonstrated effective surface and water disinfection. These findings confirm their strong antimicrobial potential.

Comment: There are numerous mistakes in terms of grammar and typos.

Response: Thank you for pointing out the grammatical and typographical errors. We have carefully revised the entire manuscript to correct errors in grammar, punctuation, and wording, thereby enhancing clarity and readability throughout the text.

Comment: Please report results in the conclusion section. What are the future aspects of the research work conducted?

Response: Thank you for your insightful comment. We have revised the Conclusion section to include key antimicrobial results, such as MIC values, inhibition zones, and surface disinfection efficacy. We have also outlined future directions, including formulation development, synergy with antibiotics, clinical applications, and assessments of biocompatibility and scalability.

Reviewer #3

Comment: In zinc nanopartical sysnthesis in formula acetate spelling correction needed

Response:

---

## [Decision Letter · Decision Letter 1]

13 May 2025

We look forward to receiving your revised manuscript.

Kind regards,

Hamida Hamdi Mohammed Ismail, ph.D.

Academic Editor

PLOS ONE

Journal Requirements:

**Additional Editor Comments:**

We note that one or more reviewers has recommended that you cite specific previously published works. As always, we recommend that you please review and evaluate the requested works to determine whether they are relevant and should be cited. It is not a requirement to cite these works. We appreciate your attention to this request.

Reviewers' comments:

Reviewer's Responses to Questions

**Comments to the Author**

Reviewer #2: (No Response)

Reviewer #5: All comments have been addressed

2. Is the manuscript technically sound, and do the data support the conclusions?

Reviewer #2: Yes

Reviewer #5: Yes

3. Has the statistical analysis been performed appropriately and rigorously?

Reviewer #2: Yes

Reviewer #5: Yes

4. Have the authors made all data underlying the findings in their manuscript fully available?

Reviewer #2: Yes

Reviewer #5: Yes

5. Is the manuscript presented in an intelligible fashion and written in standard English?

Reviewer #2: Yes

Reviewer #5: Yes

Reviewer #2: Although authors have addressed maximum points raised however, I would recommend incorporating following points before the publications of the article.

Recommendation: Minor Revision

1. The authors should provide more information in the figure legends for non-experts. All legends should have enough description for a reader to understand the figure without having to refer back to the main text of the manuscript. For example, the necessary expansion (for abbreviations) should be given which are used in the present investigation.

2. Update the introduction section and cite these recent literature to attract a broad readership in the area i) Low-Friction Soft Robots for Targeted Bacterial Infection Treatment in Gastrointestinal Tract, ii) Pharmacokinetics effects of chuanxiong rhizoma on warfarin in pseudo germ-free rats, iii) Self-assembly multifunctional DNA tetrahedron for efficient elimination of antibiotic-resistant bacteria

3. While discussing medicinal properties of nanoparticles cite i) Advancing Aggregation-Induced Emission-Derived Biomaterials in Viral, Tuberculosis, and Fungal Infectious Diseases, ii) Dual recombinase polymerase amplification system combined with lateral flow immunoassay for simultaneous detection of Staphylococcus aureus and Vibrio parahaemolyticus

Reviewer #5: The authors have made corrections according to suggestions, no more comments now, the manuscript may be accepted for online publication, good luck

**Do you want your identity to be public for this peer review?** For information about this choice, including consent withdrawal, please see our Privacy Policy

Reviewer #2: **Yes:** Azhar Abbas

Reviewer #5: No

---

## [Author Response · Author response to Decision Letter 2]

10 Jul 2025

please see the attached response point by point letter file name Response to reviewer round 2.

---

## [Decision Letter · Decision Letter 2]

16 Oct 2025

Dear Dr. Atoom,

Thank you for submitting your manuscript to PLOS ONE. After careful consideration, we feel that it has merit but does not fully meet PLOS ONE’s publication criteria as it currently stands. Therefore, we invite you to submit a revised version of the manuscript that addresses the points raised during the review process.

We look forward to receiving your revised manuscript.

Kind regards,

Hamida Hamdi Mohammed Ismail, ph.D.

Academic Editor

PLOS ONE

Journal Requirements:

Reviewers' comments:

Reviewer's Responses to Questions

**Comments to the Author**

Reviewer #1: (No Response)

2. Is the manuscript technically sound, and do the data support the conclusions?

Reviewer #1: Partly

3. Has the statistical analysis been performed appropriately and rigorously?

Reviewer #1: N/A

4. Have the authors made all data underlying the findings in their manuscript fully available?

Reviewer #1: No

5. Is the manuscript presented in an intelligible fashion and written in standard English?

Reviewer #1: No

Reviewer #1: Your manuscript addresses an important area of research, particularly in the context of antimicrobial resistance and the potential of nanomaterials in disinfection. The work is promising; however, several clarifications and corrections are needed to improve transparency, reproducibility, and clarity. Please consider the following comments:

Fungal assay controls

You mention the use of E. coli ATCC 25922, S. aureus ATCC 25923, and P. aeruginosa ATCC 27853 as standard quality control strains for antibacterial assays. However, it is not clear what reference strains or controls were used for the antifungal assays. Please provide this information.

Plant material collection

If plants were used in nanoparticle synthesis, the site of collection must be mentioned, along with details of botanical authentication (voucher specimen, herbarium record, or expert identification). This is important for reproducibility.

Equipment, reagents, and chemicals

Provide the make, model, and country of origin for the instruments (e.g., spectrophotometer, centrifuge, electron microscope) and for all key reagents and chemicals used in the experiments.

Correction in terminology

The statement “The antibacterial activity of synthesized ZnO-NPs was assessed against fungi” is incorrect and should be revised to “The antifungal activity of synthesized ZnO-NPs was assessed against fungi.”

Clarification of ZnO₂-NPs

The manuscript refers to ZnO₂-NPs. Please clarify whether this is a typographical error for ZnO-NPs, or if zinc peroxide nanoparticles (ZnO₂) were also synthesized and tested. Consistency in chemical nomenclature is critical.

**Do you want your identity to be public for this peer review?** For information about this choice, including consent withdrawal, please see our Privacy Policy

Reviewer #1: No

---

## [Author Response · Author response to Decision Letter 3]

28 Nov 2025

Response to reviewers

We sincerely thank the reviewer for the careful and constructive evaluation of our manuscript. We greatly appreciate the time and effort devoted to providing detailed comments and suggestions, which have helped us improve the clarity, accuracy, and overall quality of our work. We have carefully addressed each point raised, and the manuscript has been revised accordingly to reflect these improvements.

In addition, we have thoroughly revised the manuscript to improve language and clarity. All data presented in the tables are now also included in the supplementary materials, with the addition of two new supplementary files. Furthermore, the figures in the manuscript provide visual confirmation of the antimicrobial effects, as assessed using multiple complementary methodologies.

Reviewer #1: Your manuscript addresses an important area of research, particularly in the context of antimicrobial resistance and the potential of nanomaterials in disinfection. The work is promising; however, several clarifications and corrections are needed to improve transparency, reproducibility, and clarity. Please consider the following comments:

Response to Reviewer 1 comments:

Comment 1: Fungal assay controls: You mention the use of E. coli ATCC 25922, S. aureus ATCC 25923, and P. aeruginosa ATCC 27853 as standard quality control strains for antibacterial assays. However, it is not clear what reference strains or controls were used for the antifungal assays. Please provide this information.

Response to comment 1: Thank you for your valuable comment and for highlighting the need for clarification regarding the fungal reference strains used in the antifungal assays. We confirm that standard fungal isolates were used in our study, as listed in Table 1. In accordance with your valuable suggestion, we have revised the relevant paragraph in the Methods section to include the ATCC numbers of all standard fungal isolates employed: Candida albicans ATCC 90028, Candida parapsilosis ATCC 22019, and Candida glabrata ATCC 2001. Additionally, we have included Haemophilus influenzae ATCC 49247 to ensure that the description fully reflects all ATCC strains utilized in this work. These details have now been clearly incorporated into the manuscript to improve transparency and methodological clarity.

Comment 2: Plant material collection:

If plants were used in nanoparticle synthesis, the site of collection must be mentioned, along with details of botanical authentication (voucher specimen, herbarium record, or expert identification). This is important for reproducibility.

Response to comment 2: Thank you for your thoughtful comment. We appreciate the importance of providing complete information on plant material collection to ensure reproducibility. In this study, the plant material used for nanoparticle synthesis consisted of root hairs from Phoenix dactylifera. These plants were cultivated and collected from the campus of Al-Ahliyya Amman University (Amman, Jordan), ensuring a controlled and well-documented source.

The botanical identification of Phoenix dactylifera was verified by an expert botanist in the Faculty of Agricultural Technology at Al-Ahliyya Amman University, based on established morphological criteria. Although a voucher specimen was not prepared, the identification was conducted by qualified personnel, and this information has now been clearly incorporated into the revised manuscript to ensure full transparency and traceability of the plant material used.

We believe that this additional clarification fully addresses the reviewer’s concern and strengthens the methodological rigor of our study.

Comment 3: Equipment, reagents, and chemicals

Provide the make, model, and country of origin for the instruments (e.g., spectrophotometer, centrifuge, electron microscope) and for all key reagents and chemicals used in the experiments.

Response to comment 3: Thank you for your helpful comment. We agree that providing complete information on the equipment, reagents, and chemicals used in this study is essential for reproducibility and methodological transparency. In response, we have revised the Materials and Methods section to include the make, model, and country of origin for all major instruments utilized.

Additionally, we have provided full details for all key reagents and chemicals employed in the experiments, including the manufacturer and country of origin. These additions ensure that the experimental procedures can be accurately replicated and that all methodological specifications are fully traceable.

Comment 4: Correction in terminology:

The statement “The antibacterial activity of synthesized ZnO-NPs was assessed against fungi” is incorrect and should be revised to “The antifungal activity of synthesized ZnO-NPs was assessed against fungi.”

Response to comment 4: Thank you for bringing this terminology issue to our attention. We fully agree with your observation. The statement has been corrected from “The antibacterial activity of synthesized ZnO-NPs was assessed against fungi” to “The antifungal activity of synthesized ZnO-NPs was assessed against fungi.”

Comment 5: Clarification of ZnO₂-NPs:

The manuscript refers to ZnO₂-NPs. Please clarify whether this is a typographical error for ZnO-NPs, or if zinc peroxide nanoparticles (ZnO₂) were also synthesized and tested. Consistency in chemical nomenclature is critical.

Response to comment 5: Thank you for your careful reading and for pointing out the inconsistency in the nanoparticle nomenclature. We apologize for the typographical errors in the manuscript. To clarify, only zinc oxide nanoparticles (ZnO-NPs) were synthesized and tested in this study. The term “ZnO₂-NPs” appeared mistakenly in two places and does not reflect the materials used. We have carefully reviewed the entire manuscript and corrected these occurrences to ensure full consistency and accuracy in chemical nomenclature. We appreciate your attention to detail, as this correction helps improve the clarity and scientific precision of our work.

---

## [Decision Letter · Decision Letter 3]

22 Dec 2025

Analysis of the Antimicrobial Activity of Zinc Oxide Nanoparticles Against Drug-Resistant Bacteria and Their Applications in the Disinfection Process

PONE-D-24-57702R3

Dear Dr. Atoom,

We’re pleased to inform you that your manuscript has been judged scientifically suitable for publication and will be formally accepted for publication once it meets all outstanding technical requirements.

Kind regards,

Hamida Hamdi Mohammed Ismail, ph.D.

Academic Editor

PLOS One

Additional Editor Comments (optional):

Reviewers' comments:

Reviewer's Responses to Questions

**Comments to the Author**

Reviewer #1: All comments have been addressed

2. Is the manuscript technically sound, and do the data support the conclusions?

Reviewer #1: Yes

3. Has the statistical analysis been performed appropriately and rigorously?

Reviewer #1: I Don't Know

4. Have the authors made all data underlying the findings in their manuscript fully available?

Reviewer #1: Yes

5. Is the manuscript presented in an intelligible fashion and written in standard English?

Reviewer #1: Yes

Reviewer #1: The authors have made justifiable rebuttal to the comments raised and the manuscript can be accepted in the present form.

**Do you want your identity to be public for this peer review?** For information about this choice, including consent withdrawal, please see our Privacy Policy

Reviewer #1: **Yes:** Jess Vergis

---

## [Editor Report · Acceptance letter]

PONE-D-24-57702R3

PLOS One

Dear Dr. Atoom,

I'm pleased to inform you that your manuscript has been deemed suitable for publication in PLOS One. Congratulations! Your manuscript is now being handed over to our production team.

Kind regards,

on behalf of

Professor Hamida Hamdi Mohammed Ismail

Academic Editor

PLOS One